# Near-Optimal Reward-Free Exploration for Linear Mixture MDPs with Plug-in Solver

**Xiaoyu Chen & Jiachen Hu**
Key Laboratory of Machine Perception, MOE,
School of Artificial Intelligence, Peking University
{cxy30, NickH}@pku.edu.cn

**Lin F. Yang** *
Electrical and Computer Engineering Department,
University of California, Los Angeles
linyang@ee.ucla.edu

**Liwei Wang** *
Key Laboratory of Machine Perception, MOE,
School of Artificial Intelligence, Peking University
International Center for Machine Learning Research, Peking University
wanglw@cis.pku.edu.cn

## Abstract

Although model-based reinforcement learning (RL) approaches are considered more sample efficient, existing algorithms are usually relying on sophisticated planning algorithm to couple tightly with the model-learning procedure. Hence the learned models may lack the ability of being re-used with more specialized planners. In this paper we address this issue and provide approaches to learn an RL model efficiently without the guidance of a reward signal. In particular, we take a plug-in solver approach, where we focus on learning a model in the exploration phase and demand that *any planning algorithm* on the learned model can give a near-optimal policy. specifically, we focus on the linear mixture MDP setting, where the probability transition matrix is a (unknown) convex combination of a set of existing models. We show that, by establishing a novel exploration algorithm, the plug-in approach learns a model by taking $\tilde{O}(d^2 H^3/\epsilon^2)$ episodes with the environment and *any $\epsilon$-optimal planner* on the model gives an $O(\epsilon)$-optimal policy on the original model. This sample complexity matches our lower bound for non-plug-in approaches and is *statistically optimal*. We achieve this result by leveraging a careful maximum total-variance bound using Bernstein inequality and properties specified to linear mixture MDPs.

## 1 Introduction

In reinforcement learning, an agent repeatedly interacts with the unknown environment in order to maximize the cumulative reward. To achieve this goal, an RL algorithm must be equipped with effective exploration mechanisms to learn the unknown environment and find a near-optimal policy. Efficient exploration is critical to the success of reinforcement learning algorithms, which has been widely investigated from both the empirical and the theoretical perspectives (e.g. Stadie et al. (2015); Pathak et al. (2017); Azar et al. (2017); Jin et al. (2018)). Model-based RL is one of the important approaches to solve for the RL environment. In model-based RL, the agent learns the model of the environment and then performs planning in the estimated model. It has been widely applied in many RL scenarios, including both online setting (Kaiser et al., 2019; Luo et al., 2019; Azar et al., 2017) and offline setting (Yu et al., 2020; Kidambi et al., 2020). It is also believed that model-based RL is significantly more sample-efficient than model-free RL, which has been justified by many recent empirical results (e.g. Kaiser et al. (2019); Wang et al. (2019)). Though the theoretical model-based learning in small scale problems has been studied extensively (Azar et al., 2017; Zhou et al., 2020a; Jin et al., 2020a), it is still far from complete, especially with the presence of a function approximator.

---

*Corresponding author.

As an important implication of model-based approaches, the power of *plug-in* approach have been studied in several works (Cui & Yang, 2020; Agarwal et al., 2020). The idea of plug-in approach is rather simple: We construct an empirical Markov Decision Process (MDP) using maximum likelihood estimate, then return the (approximate) optimal policy with *efficient planning algorithm* in this empirical model. The significance of plug-in approaches is two-folded. For one thing, it preserves an empirical model that keeps the value of the policies, which is of independent interests. For another, the empirical model can be used for any down-stream tasks, which makes the application much more flexible. It is shown that the plug-in approach achieves the minimax sample complexity to compute the $\epsilon$-optimal policies with a generative model in the tabular (Agarwal et al., 2020) and linear settings (Cui & Yang, 2020).

In this paper, we aim to understand the power of plug-in approach in the reward-free exploration with linear function approximation. We study the linear mixture MDPs, where the transition probability kernel is a linear mixture of a number of basis kernels (Ayoub et al., 2020; Zhou et al., 2020a;b). We first build an empirical model with an estimation of the transition dynamics in the exploration phase, and then find a near-optimal policy by planning with the empirical model via arbitrary plug-in solver in the planning phase. Our setting is different from the reward-free exploration with linear function approximation without plug-in model (Wang et al., 2020; Zanette et al., 2020b), in which the agent can directly observe all history samples and design specialized model-free algorithm in the planning phase.

Our results show that the plug-in approach can achieve near-optimal sample complexity in the reward-free setting. In particular, we proposed a statistically efficient algorithm for reward-free exploration. Our algorithm samples $\tilde{O}(d^2 H^4/\epsilon^2)$ trajectories during the exploration phase, which suffices to obtain $O(\epsilon)$-optimal policies for an arbitrary reward function with an $\epsilon$-optimal pluging solver in the planning phase. Here $d$ is the feature dimension, and $H$ is the planning horizon. Furthermore, with a more refined trajectory-wise uncertainty estimation, we further improve the sample complexity bound to $\tilde{O}\left(d^2 H^3/\epsilon^2\right)$ in the regime where $d > H$ and $\epsilon \leq H/\sqrt{d}$. This matches our lower bound $\Omega(d^2 H^3/\epsilon^2)$ for reward-free exploration in linear mixture MDPs, which indicates that our upper bound is near-optimal except for logarithmic factors. To the best of our knowledge, this is the first work that obtains minimax sample complexity bounds for the plug-in approach in reward-free exploration with linear function approximation.

## 2 RELATED WORK

**RL with Linear Function Approximation**   Reinforcement learning with linear function approximation has been widely studied in the recent few years (e.g. Jiang et al. (2017); Yang & Wang (2019; 2020); Jin et al. (2020b); Modi et al. (2020); Du et al. (2019); Zanette et al. (2020a); Cai et al. (2020); Ayoub et al. (2020); Weisz et al. (2021); Zhou et al. (2020a;b)). The linear mixture MDPs model studied in our work assumes the transition probability function is parameterized as a linear function of a given feature mapping over state-action-next-state triple (Ayoub et al., 2020; Zhou et al., 2020b;a). Based on the Bernstein inequality for vector-valued martingales, Zhou et al. (2020a) proposed an efficient algorithm that obtains minimax regret in the regime where $d > H$. Besides linear mixture MDPs, linear MDPs is another category of RL with linear function approximation, which assumes both the transition probability function and reward function are parameterized as a linear function of a given feature mapping over state-action pairs. The algorithms with best regret bounds were proposed by Jin et al. (2020b) and Yang & Wang (2020), which studied model-free algorithm and model-based algorithm respectively. The minimax regret bound for linear MDPs is still unclear.

**Reward-Free Reinforcement Learning**   In contrast to the standard RL setting, reward-free reinforcement learning separates the exploration problem and the planning problem, which allows one to handle them in a theoretically principled way. For tabular setting, reward-free reinforcement learning has been well-exploited in many previous results (Jin et al., 2020a; Kaufmann et al., 2021; Ménard et al., 2020; Zhang et al., 2020; 2021b; Wu et al., 2021a; Bai & Jin, 2020; Liu et al., 2021), where the minimax rate is obtained by Ménard et al. (2020). For reward-free exploration with linear function approximation, Wang et al. (2020) proposed the first efficient algorithm that obtains $O(d^3 H^6/\epsilon^2)$ sample complexity for linear MDPs. However, their algorithm is model-free in nature and cannot guarantee good performance with any plug-in solver. Further, Qiu et al. (2021) proposed the first provably efficient reward-free algorithm with kernel and neural function approximation.

We also noticed that there is a concurrent work which also studied reward-free exploration for linear mixture MDPs (Zhang et al., 2021a). Compared with their results, we focus on the setting of the reward-free exploration with plug-in solver, which covers the standard reward-free setting studied in the previous results. Furthermore, our sample complexity bounds are tighter than theirs by a factor of $H^2$ [1]. The above two differences introduce new challenges in both the algorithmic design and the complexity analysis in this work, which makes our algorithms much more complicated than theirs. Besides, our lower bound is tighter than theirs in the dependence on $d$.

**Plug-in Approach**   The plug-in approach has been studied in tabular/linear case in restrictive settings. E.g., Agarwal et al. (2020) and Cui & Yang (2020) studied the standard plug-in approach with a generative model, where the algorithm is allowed to query the outcome of any state action pair from an oracle. They showed the plug-in approach also achieved the minimax optimal sample complexity to find an $\epsilon$-optimal policy in both tabular MDPs and linear MDPs. The reward-free algorithms proposed by Jin et al. (2020a) are model-based in nature, thus can be regarded as a solution in the plug-in solver setting. However, their algorithms are restricted to the tabular case and cannot be applied to the setting with linear function approximation.

## 3   PRELIMINARIES

### 3.1   EPISODIC MDPS

We consider the setting of episodic Markov decision processes (MDPs), which can be denoted by a six-tuple $(\mathcal{S}, \mathcal{A}, P, R, H, \nu)$, where $\mathcal{S}$ is the set of states, $\mathcal{A}$ is the action set, $P$ is the transition probability matrix so that $P_h(\cdot|s, a)$ gives the distribution over states if action $a$ is taken on state $s$ at step $h$, $R_h(s, a)$ is the deterministic reward function of taking action $a$ on state $s$ with support $[0, 1]$ in step $h$, $H$ is the number of steps in each episode, and $\nu$ is the distribution of the initial state.

In episode $k$, the agent starts from an initial state $s_{k,1}$ sampled from the distribution $\nu$. At each step $h \in [H]$, the agent observes the current state $s_{k,h} \in \mathcal{S}$, takes action $a_{k,h} \in \mathcal{A}$, receives reward $R_h(s_{k,h}, a_{k,h})$, and transits to state $s_{k,h+1}$ with probability $P_h(s_{k,h+1}|s_{k,h}, a_{k,h})$. The episode ends when $s_{H+1}$ is reached.

A deterministic policy $\pi$ is a collection of $H$ policy functions $\{\pi_h : \mathcal{S} \to \mathcal{A}\}_{h \in [H]}$. We use $\Pi$ to denote the set of all deterministic policies. For a specific reward function $R$, we use $V_h^\pi : \mathcal{S} \times R \to \mathbb{R}$ to denote the value function at step $h$ under policy $\pi$ w.r.t. reward $R$, which gives the expected sum of the remaining rewards received under policy $\pi$ starting from $s_h = s$, i.e. $V_h^\pi(s, R) = \mathbb{E}\left[\sum_{h'=h}^H R(s_{h'}, \pi_{h'}(s_{h'})) \mid s_h = s, P\right]$. Accordingly, we define $Q_h^\pi(s, a, R)$ as the expected Q-value function at step $h$: $Q_h^\pi(s, a, R) = \mathbb{E}\left[R(s_h, a_h) + \sum_{h'=h+1}^H R(s_{h'}, \pi_{h'}(s_{h'})) \mid s_h = s, a_h = a, P\right]$.

We use $\pi_R^*$ to denote the optimal policy w.r.t. reward R, and we use $V_h^*(\cdot, R)$ and $Q_h^*(\cdot, \cdot, R)$ to denote the optimal value and Q-function under optimal policy $\pi_R^*$ at step $h$. We say a policy $\pi$ is $\epsilon$-optimal w.r.t. reward $R$ if $\mathbb{E}\left[\sum_{h=1}^H R_h(s_h, a_h) \mid \pi\right] \geq \mathbb{E}\left[\sum_{h=1}^H R_h(s_h, a_h) \mid \pi_R^*\right] - \epsilon$.

For the convenience of explanation, we assume the agent always starts from the same state $s_1$ in each episode. It is straightforward to extend to the case with stochastic initialization, by adding a initial state $s_0$ with no rewards and only one action $a_0$, and the transition probability of $(s_0, a_0)$ is the initial distribution $\mu$. We use $P_h V(s, a, R)$ as a shorthand of $\sum_{s'} P_h(s'|s, a) V(s', R)$.

### 3.2   LINEAR MIXTURE MDPS

We study a special class of MDPs called linear mixture MDPs, where the transition probability kernel is a linear mixture of a number of basis kernels (Ayoub et al., 2020; Zhou et al., 2020b;a). This model is defined as follows in the previous literature.

---

[1] When transformed to the time-homogeneous MDPs setting studied in Zhang et al. (2021a), our algorithms can achieve sample complexity bounds $\tilde{O}(d^2 H^3/\epsilon^2)$ and $\tilde{O}((d^2 H^2 + dH^3)/\epsilon^2)$, respectively.

**Definition 1.** *(Ayoub et al., 2020) Let $\phi(s, a, s') : \mathcal{S} \times \mathcal{A} \times \mathcal{S} \to \mathbb{R}^d$ be a feature mapping satisfying that for any bounded function $V : \mathcal{S} \to [0, 1]$ and any tuple $(s, a) \in \mathcal{S} \times \mathcal{A}$, we have $\|\phi_V(s, a)\|_2 \leq 1$, where $\phi_V(s, a) = \sum_{s' \in \mathcal{S}} \phi(s, a, s') V(s')$. An MDP is called a linear mixture MDP if there exists a parameter vector $\theta_h \in \mathbb{R}^d$ with $\|\theta_h\|_2 \leq B$ for a constant $B$ and a feature vector $\phi(\cdot, \cdot, \cdot)$, such that $P_h(s'|s, a) = \theta_h^\top \phi(s, a, s')$ for any state-action-next-state triplet $(s, a, s') \in \mathcal{S} \times \mathcal{A} \times \mathcal{S}$ and step $h \in [H]$.*

### 3.3 REWARD-FREE REINFORCEMENT LEARNING

We study the problem of reward-free exploration with plug-in solver. Our setting is different from the reward-free exploration setting studied in the previous literature. Formally, there are two phases in this setting: exploration phase and planning phase.

During the exploration phase, the agent interacts with the environment for $K$ episodes. In episode $k$, the agent chooses a policy $\pi_k$ which induces a trajectory. The agent observes the states and actions $s_{k,1}, a_{k,1}, \cdots, s_{k,H}, a_{k,H}$ as usual, but does not observe any rewards. After $K$ episodes, the agent calculates the estimated model $\{\tilde{P}_h = \tilde{\theta}_h^\top \phi\}_{h \in [H]}$, which will be used in the planning phase to calculate the optimal policy.

During the planning phase, the agent is no longer allowed to interact with the MDP. Also, it cannot directly observe the history samples obtained in the exploration phase. Instead, the agent is given a set of reward function $\{R_h\}_{h \in [H]}$, where $R_h : \mathcal{S} \times \mathcal{A} \to [0, 1]$ is the deterministic reward in step $h$. For notation convenience, we occasionally use $R$ as a shorthand of $\{R_h\}_{h \in [H]}$ during the analysis. We define $\hat{V}_h^{\pi, \tilde{P}}(s, R)$ as the value function of transition $\tilde{P}$ and reward $R$, i.e. $\hat{V}_h^{\pi, \tilde{P}}(s, R) = \mathbb{E}\left[\sum_{h'=h}^H R_h(s_{h'}, \pi_{h'}(s_{h'})) \mid s_h = s, \tilde{P}\right]$.

In the planning phase, the agent calculates the optimal policy $\hat{\pi}_R$ with respect to the reward function $R$ in the estimated model $\tilde{P}$ using any $\epsilon_{\mathrm{opt}}$-optimal model-based solver. That is, the returned policy $\hat{\pi}_R$ satisfies: $\hat{V}_1^{*, \tilde{P}}(s_1, R) - \hat{V}_1^{\hat{\pi}_R, \tilde{P}}(s_1, R) \leq \epsilon_{\mathrm{opt}}$.

The agent's goal is to output an accurate model estimation $\tilde{P}$ after the exploration phase, so that the policy $\hat{\pi}_R$ calculated in planning phase can be $\epsilon + \epsilon_{\mathrm{opt}}$-optimal w.r.t. any reward function $R$. Compared with the reward-free setting studied in the previous literature (Jin et al., 2020a; Wang et al., 2020; Ménard et al., 2020; Kaufmann et al., 2021), the main difference is that we require that the algorithm maintains a model estimation instead of all the history samples after the exploration phase, and can use any model-based solver to calculate the near-optimal policy in the planning phase.

## 4 REWARD-FREE RL WITH PLUG-IN SOLVER

### 4.1 ALGORITHM

The exploration phase of the algorithm is presented in Algorithm 1. Recall that for a given value function $\tilde{V}_{k,h+1}$, we have $P_h \tilde{V}_{k,h+1}(s_{k,h}, a_{k,h}) = \theta_h^\top \phi_{\tilde{V}_{k,h+1}}(s_{k,h}, a_{k,h})$ for any $k, h$. Therefore, $\tilde{V}_{k,h+1}(s_{k,h+1})$ and $\phi_{\tilde{V}_{k,h+1}}(s_{k,h}, a_{k,h})$ can be regarded as the stochastic reward and the linear feature of a linear bandits problem with linear parameter $\theta_h$. We employ the standard least-square regression to learn the underlying parameter $\theta_h$. In each episode, we first update the estimation $\hat{\theta}_k$ based on history samples till episode $k - 1$. We define the auxiliary rewards $R_{k,h}$ to guide exploration. We calculate the optimistic Q-function using the parameter estimation $\hat{\theta}_k$, and then execute the greedy policy with respect to the updated Q-function to collect new samples.

The main problem is how to define the exploration-driven reward $R_{k,h}(s, a)$, which measures the uncertainty for state-action pair $(s, a)$ at the current step. In the setting of linear mixture MDPs, the linear feature $\phi_V(s, a)$ is a function of both state-action pair $(s, a)$ and the next-step value function $V$. This is not a big deal in the standard exploration setting (Ayoub et al., 2020; Zhou et al., 2020a). However, in the reward-free setting, since the reward function is not given beforehand, we need to upper bound the estimation error of value functions for *any* possible rewards. To tackle

---

**Algorithm 1** Reward-free Exploration: Exploration Phase

---

    Input: Failure probability $\delta > 0$ and target accuracy $\epsilon > 0$
    $\lambda \leftarrow B^{-2}, \beta \leftarrow H\sqrt{d\log(4H^3K\lambda^{-1}\delta^{-1})} + \sqrt{\lambda}B, \mathcal{V} \leftarrow \{V : \mathcal{S} \to [0, H]\}$
    **for** episode $k = 1, 2, \cdots, K$ **do**
        $Q_{k,H+1}(\cdot, \cdot) = 0, V_{k,H+1}(\cdot) = 0$
5:      **for** step $h = H, H - 1, \cdots, 1$ **do**
            $\Lambda_{k,h} \leftarrow \sum_{t=1}^{k-1} \phi_{t,h}(s_{t,h}, a_{t,h})\phi_{t,h}(s_{t,h}, a_{t,h})^\top + \lambda I$
            $\hat{\theta}_{k,h} \leftarrow (\Lambda_{k,h})^{-1} \sum_{t=1}^{k-1} \phi_{t,h}(s_{t,h}, a_{t,h})\tilde{V}_{t,h+1,s,a}(s_{t,h+1})$
            $\tilde{V}_{k,h+1,s,a} \leftarrow \arg\max_{V \in \mathcal{V}} \left\|\sum_{s'} \phi(s, a, s')V(s')\right\|_{(\Lambda_{k,h})^{-1}}, \forall s, a$
            $\phi_{k,h}(s, a) \leftarrow \sum_{s'} \phi(s, a, s')\tilde{V}_{k,h+1,s,a}(s'), \forall s, a$
10:        $u_{k,h}(s, a) \leftarrow \beta\sqrt{\phi_{k,h}(s, a)^\top (\Lambda_{k,h})^{-1} \phi_{k,h}(s, a)}, \forall s, a$
            Define the exploration-driven reward function $R_{k,h}(s, a) = u_{k,h}(s, a), \forall s, a$
            $Q_{k,h}(s, a) \leftarrow \min\left\{\hat{\theta}_{k,h}^\top \left(\sum_{s'} \phi(s, a, s')V_{k,h+1}(s')\right) + R_{k,h}(s, a) + u_{k,h}(s, a), H\right\}$
            $V_{k,h}(s) \leftarrow \max_{a \in \mathcal{A}} Q_{k,h}(s, a), \pi_{k,h}(s) = \arg\max_{a \in \mathcal{A}} Q_{k,h}(s, a)$
      **end for**
15:     **for** step $h = 1, 2, \cdots, H$ **do**
        Take action $a_{k,h} = \pi_{k,h}(s_{k,h})$ and observe $s_{k,h+1} \sim P_h(s_{k,h}, a_{k,h})$
      **end for**
    **end for**
    Find $\tilde{P}_h$ such that the transition $\tilde{P}_h(\cdot|\cdot, \cdot) = \tilde{\theta}_h^\top \phi(\cdot, \cdot, \cdot)$ is well-defined and $\left\|\tilde{\theta}_h - \hat{\theta}_{K,h}\right\|_{\Lambda_{K,h}} \leq \beta$ for $h \in [H]$
20: Output: $\{\tilde{P}_h\}_{h=1}^{H}$

---

this problem, we use $\max_{V \in \mathcal{V}} \beta\|\phi_V(s, a)\|_{\Lambda_{k,h}^{-1}} = \max_{V \in \mathcal{V}} \beta\sqrt{\phi_V(s, a)^\top (\Lambda_{k,h})^{-1} \phi_V(s, a)}$ as a measure of the maximum uncertainty for the state-action pair $(s, a)$, where $\mathcal{V} = \{V : \mathcal{S} \to [0, H]\}$ is the set of all possible value functions, and $\Lambda_{k,h}$ is the summation of all the history samples $\{s_{t,h}, a_{t,h}, s_{t,h+1}\}_{t=1}^{k-1}$ with feature $\phi_{t,h}(s_{t,h}, a_{t,h}) = \arg\max_{\phi_V} \beta\|\phi_V(s, a)\|_{\Lambda_{t,h}^{-1}}$. In each episode, we define $R_{k,h}(s, a) = u_{k,h}(s, a) = \max_{V \in \mathcal{V}} \beta\|\phi_V(s, a)\|_{\Lambda_{k,h}^{-1}}$, where $R_{k,h}(s, a)$ is the exploration-driven reward used to guide exploration, and $u_{k,h}(s, a)$ is the additional bonus term which helps to guarantee that $Q_{k,h}(s, a)$ is an optimistic estimation. Finally, the algorithm returns the model estimation $\{\tilde{P}_h\}_h$ that is well defined (i.e. $\sum_{s'} \tilde{P}_h(s'|s, a) = 1, \tilde{P}_h(s'|s, a) \geq 0, \forall s, a, s'$), and satisfy the constraints $\left\|\tilde{\theta}_h - \hat{\theta}_{K,h}\right\|_{\Lambda_{K,h}} \leq \beta$.

## 4.2 Implementation Details

Algorithm 1 involves two optimization problems in line 8 and line 19. These problems can be formulated as the standard convex optimization problem with a slight modification. Specifically, the optimization problem in line 8 of Algorithm 1 can be formulated in the following way:

$$\max_V \left\|\sum_{s'} \phi(s, a, s')V(s')\right\|_{(\Lambda_{k,h})^{-1}} \quad s.t. \quad 0 \leq V(s) \leq H, \forall s \in \mathcal{S} \tag{1}$$

In general, solving this optimization problem is hard. For the case of finite state space ($S \leq \infty$), a recent work of Zhang et al. (2021a) relaxed the problem to the following linear programming problem:

$$\max_{\mathbf{f}} \left\|\Sigma_{1,k}^{-1/2}\mathbf{\Phi}(s, a)\mathbf{f}\right\|_1 \quad s.t. \quad \|\mathbf{f}\|_\infty \leq H, \tag{2}$$

where $\mathbf{\Phi}(s, a) = \left(\phi(s, a, S_1), \cdots, \phi(s, a, S_{|\mathcal{S}|})\right)$ and $\mathbf{f} = \left(f(S_1), \cdots, f(S_{|\mathcal{S}|})\right)^\top$. As discussed in Zhang et al. (2021a), the sample complexity will be worse by a factor of $d$ if we solve the

linear programming problem as an approximation. For the case where the state apace is infinite, we can use state aggregation methods (Ren & Krogh, 2002; Singh et al., 1995) to reduce the infinite state space to finite state space and apply the approximation approaches to solve it.

The optimization problem in line 19 of Algorithm 1 is to find parameter $\tilde{\theta}_h$ satisfying several constraints. For the case where the state space is finite, we can solve this problem in the following way:

$$\min_{\tilde{\theta}_h} \left\| \tilde{\theta}_h - \hat{\theta}_{K,h} \right\|_{\Lambda_{K,h}}^2 \quad s.t. \quad \sum_{s'} \tilde{\theta}^\top \phi(s,a,s') = 1, \tilde{\theta}^\top \phi(s,a,s') \geq 0, \forall s, a \in \mathcal{S} \times \mathcal{A}$$

The above problem can be regarded as a quadratic programming problem and can be solved efficiently by the standard optimization methods. By Lemma 4, we know that the true parameter $\theta_h$ satisfies $\|\hat{\theta}_{K,h} - \theta_h\|_{\Lambda_{K,h}} \leq \beta$ with high probability. Therefore, the solution $\tilde{\theta}_h$ satisfies the constraint $\left\| \tilde{\theta}_h - \hat{\theta}_{K,h} \right\|_{\Lambda_{K,h}} \leq \beta$ with high probability.

For the case where the state space is infinite, we can also solve the above problem using state aggregation methods. In particular, if the linear mixture MDP model can be regarded as a linear combination of several base MDP models (i.e. $\phi(s,a,s') = (P_1(s'|s,a), P_2(s'|s,a), \cdots, P_d(s'|s,a))^\top$ where $P_i(s'|s,a)$ is the transition probability of certain MDP model), then we can formulate the optimization problem in the following way:

$$\min_{\tilde{\theta}_h} \left\| \tilde{\theta}_h - \hat{\theta}_{K,h} \right\|_{\Lambda_{K,h}}^2 \quad s.t. \quad \tilde{\theta}^\top \mathbf{1} = 1, \tilde{\theta} \succeq 0,$$

which can also be solved efficiently in the case of infinite state space.

### 4.3 REGRET

**Theorem 1.** *With probability at least $1 - \delta$, after collecting $K = \tilde{O}\left(\frac{d^2 H^4}{\epsilon^2}\right)$ trajectories , Algorithm 1 returns a transition model $\tilde{P}$, then for any given reward in the planning phase, a policy returned by any $\epsilon_{\mathrm{opt}}$-optimal plug-in solver on $(S, A, \tilde{P}, R, H, \nu)$ is $O(\epsilon + \epsilon_{\mathrm{opt}})$-optimal for the true MDP, $(S, A, P, R, H, \nu)$.*

We also propose a lower bound for reward-free exploration in Appendix C. Our lower bound indicates that $\Omega(d^2 H^3/\epsilon^2)$ episodes are necessary to find an $\epsilon$-optimal policy with constant probability. This lower bound is achieved by connecting the sample complexity lower bound with the regret lower bound of certain constructed learning algorithms in the standard online exploration setting. Compared with this bound, our result matches the lower bound w.r.t. the dimension $d$ and the precision $\epsilon$ except for logarithmic factors.

There is also a recent paper of Wang et al. (2020) studying reward-free exploration in the setting of linear MDPs. Their sample complexity is $\tilde{O}(d^3 H^6/\epsilon^2)$, thus our bound is better than theirs by a factor of $H^2$. Though the setting is different, we find that our parameter choice and the more refined analysis is applicable to their setting, which can help to further improve their bound by a factor of $H^2$. Please see Appendix D for the detailed discussion.

## 5 IMPROVING THE DEPENDENCE ON $H$

In this section, we close the gap on $H$ with a maximum total-variance bound using Bernstein inequality. In the previous results studying regert minimization in online RL setting (Azar et al., 2017; Zhou et al., 2020a; Wu et al., 2021b), one commonly-used approach to obtain the minimax rate is to upper bound the regret using the total variance of the value function by the Bernstein's concentration inequalities, and finally bound the summation of the one-step transition variance by the law of total variance (Lattimore & Hutter, 2012; Azar et al., 2013; 2017). However, the situation becomes much more complicated in the reward-free setting with linear function approximation. Recall that $\tilde{V}_{k,h+1,s,a}(s')$ defined in Line 9 of Algorithm 1 is the next-step value function that maximizes the uncertainty $\|\phi_V(s,a)\|_{\Lambda_{k,h}^{-1}}$ for state-action pair $(s,a)$. One naive approach is to still use

$\max_V \|\phi_V(s,a)\|_{\Lambda_{k,h}^{-1}}$ as the uncertainty measure for $(s,a)$ and upper bound the error rate by the summation of the one-step transition variance of $\tilde{V}_{k,h+1,s_{k,h},a_{k,h}}(s')$. However, we can not upper bound the variance summation of $\tilde{V}_{k,h+1,s_{k,h},a_{k,h}}(s')$ in $H$ steps by $O(H^2)$ similarly by the law of total variance, since $\{\tilde{V}_{k,h,s_{k,h},a_{k,h}}\}_{h\in[H]}$ is not the value functions induced from the same policy and transition dynamics. To tackle the above problem, we need to define the exploration-driven reward and the confidence bonus for each state-action pair in a more refined way. Our basic idea is to carefully measure the expected total uncertainty along the whole trajectories w.r.t. each MDPs in the confidence set. We will explain the detail in the following subsections.

## 5.1 Algorithm

---

**Algorithm 2** Reward-free Exploration: Exploration Phase

---

Input: Failure probability $\delta > 0$ and target accuracy $\epsilon > 0$

$\lambda \leftarrow B^{-2}, \hat{\beta} \leftarrow 16\sqrt{d\log(1 + KH^2/(d\lambda))\log(32K^2H/\delta)} + \sqrt{\lambda}B$

$\check{\beta} \leftarrow 16d\sqrt{\log(1 + KH^2/(d\lambda))\log(32K^2H/\delta)} + \sqrt{\lambda}B$

$\tilde{\beta} \leftarrow 16H^2\sqrt{d\log(1 + KH^4/(d\lambda))\log(32K^2H/\delta)} + \sqrt{\lambda}B$

5: Set $\Lambda_{i,k,h} \leftarrow \lambda I, \hat{\theta}_{i,k,h} \leftarrow \mathbf{0}$ for $k = 1, h \in [H], i = 1,2,3,4,5$

Set $\mathcal{U}_{1,h}$ to be the set containing all the $\tilde{\theta}_h$ that makes $\tilde{P}_h$ well-defined, $h \in [H]$.

**for** episode $k = 1, 2, \cdots, K$ **do**

Calculate $\pi_k, \tilde{\theta}_k, R_k = \arg\max_{\pi, \tilde{\theta}_h \in \mathcal{U}_{k,h}, R} V_{k,1}^{\pi, \tilde{P}}(s_1, R)$, where $V$ is defined in Eqn 7.

**for** step $h = 1, 2, \cdots, H$ **do**

10:  Take action according to the policy $\pi_{k,h}$ and observe $s_{k,h+1} \sim P_h(\cdot|s_{k,h}, a_{k,h})$

**end for**

**for** step $h = 1, 2, \cdots, H$ **do**

Update $\{\Lambda_{i,k+1,h}\}_{i=1}^5$ using Eqn 67, 68, 71, 72 and 77

Update the model estimation $\{\hat{\theta}_{i,k+1,h}\}_{i=1}^5$ using Eqn 69, 70, 73, 74 and 78

15:  Add the constraints (Eqn 3) to the confidence set $\mathcal{U}_{k,h}$, and obtain $\mathcal{U}_{k+1,h}$

**end for**

**end for**

Output: $\left\{\tilde{P}_{K,h}(\cdot|\cdot,\cdot) = \tilde{\theta}_{K,h}^\top \phi(\cdot,\cdot,\cdot)\right\}_{h=1}^H$.

---

Our algorithm is described in Algorithm 2. At a high level, Algorithm 2 maintains a high-confidence set $\mathcal{U}_{k,h}$ for the real parameter $\theta_h$ in each episode $k$, and calculates an optimistic value function $V_{k,1}^{\pi,\tilde{P}}(s_1, R)$ for any reward function $R$ and transition $\tilde{P}_h = \tilde{\theta}_h^\top \phi$ with $\tilde{\theta}_h \in \mathcal{U}_{k,h}$. Roughly speaking, the value function $V_{k,1}^{\pi,\tilde{P}}(s_1, R)$ measures the expected uncertainty along the whole trajectory induced by policy $\pi$ in the MDP with transition $\tilde{P}$ and reward $R$. To collect more "informative" samples and minimize the worst-case uncertainty over all possible transition dynamics and reward function, we calculate $\pi_k = \arg\max_\pi \max_{\tilde{\theta}_h \in \mathcal{U}_{k,h}, R} V_{k,1}^{\pi,\tilde{P}}(s_1, R)$, and execute the policy $\pi_k$ to collect more data in episode $k$. To ensure that the model estimation $\tilde{\theta}_{k+1,h}$ is close to the true model $\theta_h$ w.r.t. features $\phi_V(s,a)$ of different value functions $V$, we use the samples collected so far to calculate five model estimation $\{\hat{\theta}_{i,k,h}\}_{i=1}^5$ and the corresponding constraints at the end of the episode $k$. Each constraint is an ellipsoid in the parameter space centered at the parameter estimation $\hat{\theta}_{i,k,h}$ with covariance matrix $\Lambda_{i,k,h}$ and radius $\beta_i$, i.e.

$$\|\tilde{\theta}_h - \hat{\theta}_{i,k,h}\|_{\Lambda_{i,k,h}} \leq \beta_i. \tag{3}$$

We update $\mathcal{U}_{k+1,h}$ by adding these constraints to the confidence set $\mathcal{U}_{k,h}$.

The remaining problems are how to define the value function $V_{k,1}^{\pi,\tilde{P}}(s_1, R)$ that represents the expected uncertainty for policy $\pi$, transition $\tilde{P}$ and reward $R$, and how to update the model estimation $\theta_{i,k,h}$ and the confidence set $\mathcal{U}_{k,h}$.

**Uncertainty Measure** Instead of using $\max_V \|\phi_V(s,a)\|_{\Lambda_{k,h}^{-1}}$ to measure the maximum uncertainty for the state-action pair $(s,a)$ in Algorithm 1, we separately define the uncertainty along the trajectories induced by different policy $\pi$, reward function $R$ and transition dynamics $\tilde{P}_h = \tilde{\theta}_h^\top \phi$. Specifically, Recall that $\hat{V}_h^{\pi,\tilde{P}}(s,R)$ is the value function of policy $\pi$ in the MDP model with transition $\tilde{P}$ and reward $R$. We define the following exploration-driven reward for transition $\tilde{P}$, reward $R$ and policy $\pi$:

$$u_{1,k,h}^{\pi,\tilde{P}}(s,a,R) = \hat{\beta} \left\| \sum_{s'} \phi(s,a,s') \hat{V}_{h+1}^{\pi,\tilde{P}}(s',R) \right\|_{(\Lambda_{1,k,h})^{-1}}. \tag{4}$$

Suppose $\tilde{V}_{k,H+1}^{\pi,\tilde{P}}(s,R) = 0, \forall s \in \mathcal{S}$, we define the corresponding value function recursively from step $H+1$ to step 1. That is,

$$\tilde{V}_{k,h}^{\pi,\tilde{P}}(s,R) = \min\left\{ u_{1,k,h}^{\pi,\tilde{P}}(s,\pi_h(s),R) + P_h \tilde{V}_{k,h+1}^{\pi,\tilde{P}}(s,\pi_h(s),R), H \right\}, \forall s \in \mathcal{S}. \tag{5}$$

$\tilde{V}_{k,h}^{\pi,\tilde{P}}(s,R)$ can be regarded as the expected uncertainty along the trajectories induced by policy $\pi$ for the value function $\hat{V}_h^{\pi,\tilde{P}}(s,R)$. In each episode $k$, we wish to collect samples by executing the policy $\pi$ that maximizes the uncertainty measure $\max_{\tilde{\theta} \in \mathcal{U}_{k,h},R} \tilde{V}_{k,1}^{\pi,\tilde{P}}(s,R)$. However, the definition of $\tilde{V}_{k,1}^{\pi,\tilde{P}}(s,R)$ explicitly depends on the real transition dynamics $P$, which is unknown to the agent. To solve this problem, we construct an optimistic estimation of $\tilde{V}_{k,h}^{\pi,\tilde{P}}(s,R)$. We define $V_{k,h}^{\pi,\tilde{P}}(s,R)$ from step $H+1$ to step 1 recursively. Suppose $V_{k,H+1}^{\pi,\tilde{P}}(s,R) = 0$. We calculate

$$u_{2,k,h}^{\pi,\tilde{P}}(s,a,R) = \hat{\beta} \left\| \sum_{s'} \phi(s,a,s') V_{k,h+1}^{\pi,\tilde{P}}(s',R) \right\|_{(\Lambda_{2,k,h})^{-1}}, \tag{6}$$

$$V_{k,h}^{\pi,\tilde{P}}(s,R) = \min\left\{ u_{1,k,h}^{\pi,\tilde{P}}(s,\pi_h(s),R) + u_{2,k,h}^{\pi,\tilde{P}}(s,\pi_h(s),R) + \tilde{P}_h V_{k,h+1}^{\pi,\tilde{P}}(s,\pi_h(s),R), H \right\}, \tag{7}$$

where $u_{2,k,h}^{\pi,\tilde{P}}$ is the confidence bonus which ensures that the optimism $V_{k,h}^{\pi,\tilde{P}}(s,R) \geq \tilde{V}_{k,h}^{\pi,\tilde{P}}(s,R)$ holds with high probability. After calculating $V_{k,h}^{\pi,\tilde{P}}(s,R)$, we take maximization over all $\tilde{\theta}_h \in \mathcal{U}_{k,h}$ and reward function $R$, and calculate $\pi_k = \arg\max_\pi \max_{\tilde{\theta}_h \in \mathcal{U}_{k,h},R} V_{k,1}^{\pi,\tilde{P}}(s_1,R)$. We execute $\pi_k$ to collect more samples in episode $k$.

**Weighted Ridge Regression and Confidence Set** In Algorithm 2, we maintain five model estimation $\{\hat{\theta}_{i,k,h}\}_{i=1}^5$ and add five constraints to the confidence set $\mathcal{U}_{k,h}$ in each episode. These constraints can be roughly classified into three categories. The first and the third constraints are applied to ensure that $\tilde{P}_{k,h} \hat{V}_{h+1}^{\pi,\tilde{P}}(s,a,R)$ is an accurate estimation of $P_{k,h} \hat{V}_{h+1}^{\pi,\tilde{P}}(s,a,R)$ for any reward function $R$ and $\tilde{P}_{k,h} = \tilde{\theta}_{k,h}^\top \phi$ satisfying $\tilde{\theta}_{k,h} \in \mathcal{U}_{k,h}$, while the second and the forth constraints are used to guarantee that $\tilde{P}_{k,h} V_{h+1}^{\pi,\tilde{P}}(s,a,R)$ is an accurate estimation of $P_h V_{h+1}^{\pi,\tilde{P}}(s,a,R)$ for any reward function $R$ and $\tilde{P}_{k,h} = \tilde{\theta}_{k,h}^\top \phi$ satisfying $\tilde{\theta}_{k,h} \in \mathcal{U}_{k,h}$. The last constraint is applied due to technical issue and will be explained in Appendix B.2. In this subsection, we introduce the basic idea behind the construction of the first and the third constraints. The second and the forth constraints follow the same idea but consider the different value function $V_{h+1}^{\pi,\tilde{P}}(s,a,R)$. The formal definition of the parameters $\{\hat{\theta}_{i,k,h}\}_{i=1}^5$ and the constraints are deferred to Appendix B.2.

For notation convenience, we use $V_{k,h}(s)$ as a shorthand of $\hat{V}_{k,h}^{\pi_k,\tilde{P}_k}(s,R_k)$ in this part. The construction of our confidence sets is inspired by a recent algorithm called UCRL-VTR$^+$ proposed by Zhou et al. (2020a). Recall that we use $(s_{k,h}, a_{k,h})$ to denote the state-action pair that the agent

encounters at step $h$ in episode $k$. We use the following ridge regression estimator to calculate the corresponding $\hat{\theta}_{1,k,h}$ in episode $k$:

$$\hat{\theta}_{1,k,h} = \arg\min_{\theta \in \mathbb{R}^d} \lambda \|\theta\|_2^2 + \sum_{t=1}^{k-1} \left[ \theta^\top \phi_{V_{t,h+1}/\bar{\sigma}_{1,t,h}}(s_{t,h}, a_{t,h}) - V_{t,h+1}(s_{t,h+1}, R_t)/\bar{\sigma}_{1,t,h} \right]^2, \quad (8)$$

where $\bar{\sigma}_{1,k,h}^2 = \max\left\{ H^2/d, [\bar{\mathbb{V}}_{k,h} V_{k,h+1}](s_{k,h}, a_{k,h}) + E_{1,k,h} \right\}$ is an optimistic estimation of the one-step transition variance:

$$\mathbb{V}_h V_{k,h+1}(s_{k,h}, a_{k,h}) = \mathbb{E}_{s' \sim P_h(\cdot|s_{k,h}, a_{k,h})} \left[ (V_{k,h+1}(s') - P_h V_{k,h}(s_{k,h}, a_{k,h}))^2 \right].$$

In the definition of $\bar{\sigma}_{1,k,h}^2$, $\bar{\mathbb{V}}_{k,h} V_{k,h+1}(s_{k,h}, a_{k,h})$ is an empirical estimation for the variance $\mathbb{V}_h V_{k,h+1}(s_{k,h}, a_{k,h})$, and $E_{1,k,h}$ is a bonus term defined in Eqn. 61 which ensures that $\bar{\sigma}_{k,h}$ is an optimistic estimation of $\mathbb{V}_h V_{k,h+1}(s_{k,h}, a_{k,h})$. One technical issue here is how to estimate the variance $\mathbb{V}_h V_{k,h+1}(s_{k,h}, a_{k,h})$. Recall that by definition,

$$\mathbb{V}_h V_{k,h+1}(s_{k,h}, a_{k,h}) = P_h V_{k,h+1}^2(s_{k,h}, a_{k,h}) - [P_h V_{k,h+1}(s_{k,h}, a_{k,h})]^2 \quad (9)$$

$$= \theta_h^\top \phi_{V_{k,h+1}^2}(s_{k,h}, a_{k,h}) - [\theta_h^\top \phi_{V_{k,h+1}}(s_{k,h}, a_{k,h})]^2. \quad (10)$$

We use $\tilde{\theta}_{k,h}^\top \phi_{V_{k,h+1}^2}(s_{k,h}, a_{k,h}) - \left[ \tilde{\theta}_{k,h}^\top \phi_{V_{k,h+1}}(s_{k,h}, a_{k,h}) \right]^2$ as our variance estimator $\bar{\mathbb{V}}_{k,h} V_{k,h+1}(s_{k,h}, a_{k,h})$, where $\tilde{\theta}_{k,h} \in \mathcal{U}_{k,h}$ is the parameter which maximizes the value $V_{k,1}^{\pi,\tilde{P}}(s_1, R)$ in episode $k$. To ensure that $\bar{\mathbb{V}}_{k,h} V_{k,h+1}(s_{k,h}, a_{k,h})$ is an accurate estimator, we maintain another parameter estimation $\hat{\theta}_{3,k,h}$ using history samples w.r.t. the feature $\phi_{V_{k,h+1}^2}(s_{k,h}, a_{k,h})$.

$$\hat{\theta}_{3,k,h} = \arg\min_{\theta \in \mathbb{R}^d} \lambda \|\theta\|_2^2 + \sum_{t=1}^{k-1} \left[ \theta^\top \phi_{V_{t,h+1}^2}(s_{t,h}, a_{t,h}) - V_{t,h+1}^2(s_{t,h+1}) \right]^2. \quad (11)$$

After calculating $\hat{\theta}_{1,k,h}$ and $\hat{\theta}_{3,k,h}$, we add the first and the third constraints (Eqn. 3) to the confidence set $\mathcal{U}_{k,h}$, where $\Lambda_{1,k,h}$ and $\Lambda_{3,k,h}$ is the corresponding covariance matrix of all history samples, i.e. $\Lambda_{1,k,h} = \sum_{t=1}^{k-1} \bar{\sigma}_{1,t,h}^{-2} \phi_{V_{t,h+1}}(s_{t,h}, a_{t,h}) \phi_{V_{t,h+1}}^\top(s_{t,h}, a_{t,h})$ and $\Lambda_{3,k,h} = \sum_{t=1}^{k-1} \phi_{V_{t,h+1}^2}(s_{t,h}, a_{t,h}) \phi_{V_{t,h+1}^2}^\top(s_{t,h}, a_{t,h})$.

## 5.2 REGRET

We present the regret upper bound of Algorithm 2 in Theorem 2. In the regime where $d \geq H$ and $\epsilon \leq H/\sqrt{d}$, we can obtain $\tilde{O}(d^2 H^3/\epsilon^2)$ sample complexity upper bound, which matches the sample complexity lower bound except logarithmic factors.

**Theorem 2.** *With probability at least $1 - \delta$, after collecting $K = \tilde{O}\left( \frac{d^2 H^3 + dH^4}{\epsilon^2} + \frac{d^{2.5} H^2 + d^2 H^3}{\epsilon} \right)$ trajectories, Algorithm 2 returns a transition model $\tilde{P}_K$, then for any given reward in the planning phase, a policy returned by any $\epsilon_{\text{opt}}$-optimal plug-in solver on $(S, A, \tilde{P}_K, R, H, \nu)$ is $O(\epsilon + \epsilon_{\text{opt}})$-optimal for the true MDP, $(S, A, P, R, H, \nu)$.*

## 6 CONCLUSION

This paper studies the sample complexity of plug-in solver approach for reward-free reinforcement learning. We propose a statistically efficient algorithm with sample complexity $\tilde{O}\left(d^2 H^4/\epsilon^2\right)$. We further refine the complexity by providing an another algorithm with sample complexity $\tilde{O}\left(d^2 H^3/\epsilon^2\right)$ in certain parameter regimes. To the best of our knowledge, this is the first minimax sample complexity bound for reward-free exploration with linear function approximation. As a side note, our approaches provide an efficient learning method for the RL model representation, which preserves values and policies for any other down-stream tasks (specified by different rewards).

Our sample complexity bound matches the lower bound only when $d \geq H$ and $\epsilon \leq H/\sqrt{d}$. It is unclear whether minimax rate could be obtained in a more broader parameter regimes. We plan to address this issue in the future work.

## 7 ACKNOWLEDGMENTS

Liwei Wang was supported by National Key R&D Program of China (2018YFB1402600), Exploratory Research Project of Zhejiang Lab (No. 2022RC0AN02), BJNSF (L172037), Project 2020BD006 supported by PKUBaidu Fund.

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

# A OMITTED DETAILS IN SECTION 4

## A.1 NOTATIONS

In this subsection, we summarize the notations used in Section A.

| Symbol | Explanation |
|---|---|
| $\mathcal{E}_1$ | The high-probability event for Theorem 1 |
| $s_{k,h}, a_{k,h}$ | The state and action that the agent encounters in episode $k$ and step $h$ |
| $\tilde{V}_{k,h+1,s,a}$ | The value function with maximum uncertainty: $\arg\max_{V\in\mathcal{V}}\|\sum_{s'}\phi(s,a,s')V(s')\|_{(\Lambda_{k,h})^{-1}}$ |
| $\phi_{k,h}(s,a)$ | $\sum_{s'}\phi(s,a,s')\tilde{V}_{k,h+1,s,a}(s')$ |
| $\hat{\theta}_{k,h}$ | The estimation of $\theta_h$ in episode $k$: $(\Lambda_{k,h})^{-1}\sum_{t=1}^{k-1}\phi_{t,h}(s_{t,h},a_{t,h})\tilde{V}_{t,h+1,s,a}(s_{t,h+1})$ |
| $\Lambda_{k,h}$ | The covariance matrix in $(k,h)$: $\sum_{t=1}^{k-1}\phi_{t,h}(s_{t,h},a_{t,h})\phi_{t,h}(s_{t,h},a_{t,h})^{\top}+\lambda I$ |
| $u_{k,h}(s,a)$ | The uncertainty measure: $\beta\sqrt{\phi_{k,h}(s,a)^{\top}(\Lambda_{k,h})^{-1}\phi_{k,h}(s,a)}$ |
| $R_{k,h}(s,a)$ | The exploration-driven reward which equals $u_{k,h}(s,a)$ |
| $Q_{k,h}(s,a)$ | The Q function defined in line 12 of Algorithm 1 |
| $V_{k,h}(s)$ | The value function defined in line 13 of Algorithm 1 |
| $\tilde{V}_h^*(s,R)$ | The value function defined in Eqn 12 |
| $\tilde{Q}_h^*(s,R)$ | The Q value similarly defined as $\tilde{V}_h^*(s,R)$ |
| $\tilde{V}_h^\pi(s,R)$ | The value function of policy $\pi$ similarly defined as $\tilde{V}_h^*(s,R)$ |
| $\tilde{\theta}_h$ | The parameter estimation returned at the end of the exploration phase |
| $\hat{P}_{k,h}(s'|s,a)$ | $\hat{\theta}_{k,h}^{\top}\phi(s,a,s')$ |
| $\tilde{P}_h(s'|s,a)$ | $\tilde{\theta}_h^{\top}\phi(s,a,s')$ |

## A.2 PROOF OVERVIEW

Now we briefly explain the main idea in the proof. Firstly, we introduce the value function $\tilde{V}_h^*(s,R)$, which is recursively defined from step $H+1$ to step 1:

$$\tilde{V}_{H+1}^*(s,R)=0, \forall s\in\mathcal{S} \tag{12}$$

$$\tilde{V}_h^*(s,R)=\max_{a\in\mathcal{A}}\left\{\min\left\{R_h(s,a)+P_h\tilde{V}_{h+1}^*(s,a,R),H\right\}\right\}, \forall s\in\mathcal{S}, h\in[H] \tag{13}$$

Compared with the definition of $V_h^*(s,R)$, the main difference is that we take minimization over the value and $H$ at each step. We state the following lemma, which gives an upper bound on the sub-optimality gap of $\hat{\pi}$ in the planning phase.

**Lemma 1.** *With probability at least $1-\delta$, the sub-optimality gap of the policy $\hat{\pi}_R$ for any reward function $R$ in the planning phase satisfies $V_1^*(s_1,R)-V_1^{\hat{\pi}_R}(s_1,R)\le 4\tilde{V}_1^*(s_1,R_K)+\epsilon_{\mathrm{opt}}$, where $R_K$ is the exploration-driven reward used for episode $K$ in the exploration phase.*

This lemma connects the sub-optimality gap with the value function of the auxiliary reward in episode $K$. So the remaining problem is how to upper bound $\tilde{V}_1^*(s_1,R_K)$. Since the exploration-driven reward $R_k$ is non-increasing w.r.t. $k$, it is not hard to prove that $K\tilde{V}_1^*(s_1,R_K)\le\sum_{k=1}^K\tilde{V}_1^*(s_1,R_k)$. We use the following two lemmas to upper bound $\sum_{k=1}^K\tilde{V}_1^*(s_1,R_k)$.

**Lemma 2.** *With probability at least $1-\delta$, $\tilde{V}_h^*(s,R_k)\le V_{k,h}(s)$ holds for any $(s,a)\in\mathcal{S}\times\mathcal{A}, h\in[H]$ and $k\in[K]$.*

**Lemma 3.** *With probability at least $1-\delta$,*

$$\sum_{k=1}^K V_{k,1}(s_1)\le 6H^2 d\sqrt{K\log(4H^3KB^2/\delta)\log(1+KH^2B^2/d)}.$$

With the help of the optimistic bonus term $u_{k,h}$, we can prove that the estimation value $V_{k,h}$ is always optimistic w.r.t $\tilde{V}_h^*$, which is illustrated in Lemma 2. Therefore, we have $\sum_{k=1}^K\tilde{V}_1^*(s_1,R_k)\le\sum_{k=1}^K V_{k,1}(s_1)$. By adapting the regret analysis in the standard RL setting to the reward-free setting, we can upper bound the summation of $V_{k,1}(s_1)$ in Lemma 3. Combining the above two lemmas, we derive the upper bound of $\tilde{V}_1^*(s_1,R_K)$, then we bound the sub-optimality gap of $\hat{\pi}$ by Lemma 1.

A.3    HIGH-PROBABILITY EVENTS

We firstly state the following high-probability events.

**Lemma 4.** *With probability at least $1-\delta/2$, the following inequality holds for any $k \in [K], h \in [H]$:*

$$\|\hat{\theta}_{k,h} - \theta_h\|_{\Lambda_{k,h}} \leq \beta. \tag{14}$$

*Proof.* For some step $h \in [H]$, the agent selects an action with feature $x_{k,h} = \phi_{k,h}(s_{k,h}, a_{k,h})$. The noise $\eta_{k,h} = \tilde{V}_{k,h,s_{k,h},a_{k,h}}(s') - \theta_h^\top \phi_{k,h}(s_{k,h}, a_{k,h})$ satisfies $H$-sub-Gaussian. By Lemma 20, the following inequality holds with probability $1 - \frac{\delta}{2H}$:

$$\|\hat{\theta}_{k,h} - \theta_h\|_{\Lambda_{k,h}} \leq H\sqrt{d\log(\frac{4H^3K}{\lambda\delta})} + \lambda^{1/2}B. \tag{15}$$

By taking union bound over all $h \in [H]$, we can prove the lemma.

$\square$

**Lemma 5.** *With probability at least $1 - \delta/2$, we have*

$$\sum_{k=1}^{K} \sum_{h=1}^{H} (P_h V_{k,h+1}(s_{k,h}, a_{k,h}) - V_{k,h+1}(s_{k,h+1})) \leq \sqrt{2H^3K\log(4/\delta)}. \tag{16}$$

*Proof.* This lemma follows directly by Azuma's inequality.    $\square$

During the following analysis, we denote the high-probability events defined in Lemma 4 and Lemma 5 as $\mathcal{E}_1$.

A.4    PROOF OF LEMMA 1

**Lemma 6.** *(Restatement of Lemma 1) Under event $\mathcal{E}_1$, the sub-optimality gap of the policy $\hat{\pi}_R$ for any reward function $R$ in the planning phase can be bounded by*

$$V_1^*(s_1, R) - V_1^{\hat{\pi}_R}(s_1, R) \leq 4\tilde{V}_1^*(s_1, R_K) + \epsilon_{\text{opt}}. \tag{17}$$

*Proof.*

$$V_1^*(s_1, R) - V_1^{\hat{\pi}_R}(s_1, R) \tag{18}$$

$$= \left(V_1^*(s_1, R) - \hat{V}_1^{\pi_R^*, \tilde{P}}(s_1, R)\right) + \left(\hat{V}_1^{\hat{\pi}_R, \tilde{P}}(s_1, R) - V_1^{\hat{\pi}_R}(s_1, R)\right) + \left(\hat{V}_1^{\pi_R^*, \tilde{P}}(s_1, R) - \hat{V}_1^{\hat{\pi}_R, \tilde{P}}(s_1, R)\right) \tag{19}$$

$$\leq \left(V_1^*(s_1, R) - \hat{V}_1^{\pi_R^*, \tilde{P}}(s_1, R)\right) + \left(\hat{V}_1^{\hat{\pi}_R, \tilde{P}}(s_1, R) - V_1^{\hat{\pi}_R}(s_1, R)\right) + \epsilon_{\text{opt}}. \tag{20}$$

The inequality is because that the policy $\hat{\pi}_R$ is the $\epsilon_{\text{opt}}$-optimal policy in the estimated MDP $\hat{M}$, i.e. $\hat{V}_1^{\pi_R^*, \tilde{P}}(s_1, R) \leq \hat{V}_1^{\hat{\pi}_R, \tilde{P}}(s_1, R) + \epsilon_{\text{opt}}$.

For the notation convenience, for a certain sequence $\{R_h\}_{h=1}^H$, we define the function $W_h(\{R_h\})$ recursively from step $H+1$ to step 1. Firstly, we define $W_{H+1}(\{R_h\}) = 0$. $W_h(\{R_h\})$ is calculated recursively from $W_{h+1}(\{R_h\})$:

$$W_h(\{R_h\}) = \min\{H, R_h + W_{h+1}(\{R_h\})\}. \tag{21}$$

Similarly with the definition of $\tilde{V}_h^*(s, R)$, we introduce the value function $\tilde{V}_h^\pi(s, R)$, which is recursively defined from step $H + 1$ to step 1:

$$\tilde{V}_{H+1}^\pi(s, R) = 0, \forall s \in \mathcal{S} \tag{22}$$

$$\tilde{V}_h^\pi(s, R) = \left\{\min\left\{R_h(s, \pi(s)) + P_h\tilde{V}_{h+1}^\pi(s, \pi(s), R), H\right\}\right\}, \forall s \in \mathcal{S}, h \in [H]. \tag{23}$$

We use $\text{traj} \sim (\pi, P)$ to indicate that the trajectory $\{s_h, a_h\}_{h=1}^H$ is sampled from transition $P$ with policy $\pi$. For any policy $\pi$, we have

$$\left| \mathbb{E}_{s_1 \sim \mu} \left( \hat{V}_1^{\pi, \tilde{P}}(s_1, R) - V_1^\pi(s_1) \right) \right| \tag{24}$$

$$= \left| \mathbb{E}_{\text{traj} \sim (\pi, P)} W_1 \left( \left\{ (\tilde{P}_h - P_h) \hat{V}_{h+1}^{\pi, \tilde{P}_K}(s_h, a_h, R) \right\} \right) \right| \tag{25}$$

$$= \left| \mathbb{E}_{\text{traj} \sim (\pi, P)} W_1 \left( \left\{ (\tilde{\theta}_h - \theta_h) \sum_{s'} \phi(s_h, a_h, s') \hat{V}_{h+1}^{\pi, \tilde{P}}(s', R) \right\} \right) \right| \tag{26}$$

$$\leq \mathbb{E}_{\text{traj} \sim (\pi, P)} W_1 \left( \left\{ \left\| \tilde{\theta}_h - \theta_h \right\|_{\Lambda_{K,h}} \left\| \sum_{s'} \phi(s_h, a_h, s') \hat{V}_{h+1}^{\pi, \tilde{P}}(s', R) \right\|_{(\Lambda_{K,h})^{-1}} \right\} \right) \tag{27}$$

$$\leq \mathbb{E}_{\text{traj} \sim (\pi, P)} W_1 \left( \left\{ 2\beta \left\| \sum_{s'} \phi(s_h, a_h, s') \hat{V}_{h+1}^{\pi, \tilde{P}}(s', R) \right\|_{(\Lambda_{K,h})^{-1}} \right\} \right) \tag{28}$$

$$\leq 2 \mathbb{E}_{\text{traj} \sim (\pi, P)} W_1 \left( \{ u_{K,h}(s_h, a_h) \} \right) \tag{29}$$

$$= 2 \tilde{V}_1^\pi(s_1, R_K) \tag{30}$$

$$\leq 2 \tilde{V}_1^*(s_1, R_K). \tag{31}$$

The second inequality is due to lemma 4 and the definition of $\tilde{\theta}$. Plugging this inequality back to Inq. 18, we can prove the lemma. $\qquad \square$

## A.5  PROOF OF LEMMA 2

**Lemma 7.** *(Restatement of Lemma 2) Under event $\mathcal{E}_1$, $\tilde{V}_h^*(s, R_k) \leq V_{k,h}(s)$ holds for any $(s, a) \in \mathcal{S} \times \mathcal{A}, h \in [H]$ and $k \in [K]$. .*

*Proof.* We prove the lemma by induction. Suppose $\tilde{V}_{h+1}^*(s, R_k) \leq V_{k,h+1}(s)$,

$$\tilde{Q}_h^*(s, a, R_k) - Q_{k,h}(s, a) \tag{32}$$

$$\leq -u_{k,h}(s, a) + \left( \theta_{k,h} - \hat{\theta}_{k,h} \right)^\top \cdot \sum_{s'} \phi(s, a, s') V_{k,h}(s') + P_h(\tilde{V}_{h+1}^* - V_{k,h+1})(s, a) \tag{33}$$

$$\leq -u_{k,h}(s, a) + \left( \theta_{k,h} - \hat{\theta}_{k,h} \right)^\top \cdot \sum_{s'} \phi(s, a, s') V_{k,h}(s') \tag{34}$$

$$\leq -u_{k,h}(s, a) + \left\| \theta_{k,h} - \hat{\theta}_{k,h} \right\|_{\Lambda_{k,h}} \left\| \sum_{s'} \phi(s, a, s') V_{k,h}(s') \right\|_{(\Lambda_{k,h})^{-1}} \tag{35}$$

$$\leq -u_{k,h}(s, a) + \left\| \theta_{k,h} - \hat{\theta}_{k,h} \right\|_{\Lambda_{k,h}} \max_{V \in \mathcal{V}} \left\| \sum_{s'} \phi(s, a, s') V(s') \right\|_{(\Lambda_{k,h})^{-1}} \tag{36}$$

$$\leq -u_{k,h}(s, a) + \beta \max_{V \in \mathcal{V}} \left\| \sum_{s'} \phi(s, a, s') V(s') \right\|_{(\Lambda_{k,h})^{-1}} \tag{37}$$

$$= 0. \tag{38}$$

The first inequality is due to induction condition $\tilde{V}_{h+1}^*(s, R_k) \leq V_{k,h+1}(s)$. The last inequality is due to Lemma 4. Since $\tilde{Q}_h^*(s, a) \leq Q_{k,h}(s, a, R_k)$ for any $a \in \mathcal{A}$, we have $\tilde{V}_h^*(s, R_k) \leq V_{k,h}(s)$. $\qquad \square$

## A.6  PROOF OF LEMMA 3

**Lemma 8.** *(Restatement of Lemma 3) Under event $\mathcal{E}_1$, $\sum_{k=1}^K V_{k,1}(s_{k,1}) \leq 6H^2 d \sqrt{K \log(4H^3 K B^2/\delta) \log(1 + KH^2 B^2/d)}$.*

*Proof.*

$$Q_{k,h}(s_{k,h}, a_{k,h}) \tag{39}$$

$$= \min\left\{H, R_{k,h}(s_{k,a}, a_{k,h}) + u_{k,h}(s_{k,a}, a_{k,h}) + \hat{P}_{k,h}V_{k,h+1}(s_{k,h}, a_{k,h})\right\} \tag{40}$$

$$\leq \min\left\{H, 2\beta \left\|\phi_{k,h}(s_{k,h}, a_{k,h})\right\|_{\Lambda_{k,h}^{-1}} + \hat{P}_{k,h}V_{k,h+1}(s_{k,h}, a_{k,h})\right\} \tag{41}$$

$$\leq \min\left\{H, 2\beta \left\|\phi_{k,h}(s_{k,h}, a_{k,h})\right\|_{\Lambda_{k,h}^{-1}}\right\} + \min\left\{H, \left(\hat{P}_{k,h} - P_h\right)V_{k,h+1}(s_{k,h}, a_{k,h})\right\} \tag{42}$$

$$+ \left(P_h V_{k,h+1}(s_{k,h}, a_{k,h}) - V_{k,h+1}(s_{k,h+1})\right) + V_{k,h+1}(s_{k,h+1}) \tag{43}$$

$$\leq 3\beta \min\left\{1, \left\|\phi_{k,h}(s_{k,h}, a_{k,h})\right\|_{\Lambda_{k,h}^{-1}}\right\} + \left(P_h V_{k,h+1}(s_{k,h}, a_{k,h}) - V_{k,h+1}(s_{k,h+1})\right) + V_{k,h+1}(s_{k,h+1}). \tag{44}$$

The last inequality is from

$$\left(\hat{P}_{k,h} - P_h\right)V_{k,h+1}(s_{k,h}, a_{k,h}) = (\hat{\theta}_{k,h} - \theta_h)\phi_{V_{k,h+1}}(s_{k,h}, a_{k,h}) \tag{45}$$

$$\leq \left\|\hat{\theta}_{k,h} - \theta_h\right\|_{\Lambda_{k,h}} \left\|\phi_{V_{k,h+1}}(s_{k,h}, a_{k,h})\right\|_{\Lambda_{k,h}^{-1}} \tag{46}$$

$$\leq \beta \left\|\phi_{V_{k,h+1}}(s_{k,h}, a_{k,h})\right\|_{\Lambda_{k,h}^{-1}} \tag{47}$$

$$\leq \beta \left\|\phi_{k,h}(s_{k,h}, a_{k,h})\right\|_{\Lambda_{k,h}^{-1}}. \tag{48}$$

From Inq 39, we have

$$\sum_{k=1}^{K} V_{k,1}(s_1) = \sum_{k=1}^{K}\sum_{h=1}^{H} 3\beta \min\left\{1, \left\|\phi_{k,h}(s_{k,h}, a_{k,h})\right\|_{\Lambda_{k,h}^{-1}}\right\} \tag{49}$$

$$+ \sum_{k=1}^{K}\sum_{h=1}^{H} \left(P_h V_{k,h+1}(s_{k,h}, a_{k,h}) - V_{k,h+1}(s_{k,h+1})\right). \tag{50}$$

For the first term, by Lemma 21, we have for any $h \in [H]$,

$$\sum_{k=1}^{K} \min\left\{1, \left\|\phi_{k,h}(s_{k,h}, a_{k,h})\right\|_{\Lambda_{k,h}^{-1}}\right\} \leq \sqrt{K \sum_{k=1}^{K} \min\left\{1, \left\|\phi_{k,h}(s_{k,h}, a_{k,h})\right\|_{\Lambda_{k,h}^{-1}}^2\right\}} \tag{51}$$

$$\leq \sqrt{2dK \log(1 + KH^2/(d\lambda))}. \tag{52}$$

Since $\lambda = B^{-2}$, we have

$$\sum_{k=1}^{K}\sum_{h=1}^{H} 3\beta \min\left\{1, \left\|\phi_{k,h}(s_{k,h}, a_{k,h})\right\|_{\Lambda_{k,h}^{-1}}\right\} \leq 3H^2 d\sqrt{2K \log(4H^3 K B^2/\delta) \log(1 + KH^2 B^2/d)}. \tag{53}$$

For the second term, by Lemma 5,

$$\sum_{k=1}^{K}\sum_{h=1}^{H} \left(P_h V_{k,h+1}(s_{k,h}, a_{k,h}) - V_{k,h+1}(s_{k,h+1})\right) \leq \sqrt{2H^3 K \log(4/\delta)}. \tag{54}$$

Therefore, we have

$$\sum_{k=1}^{K} V_{k,1}(s_{k,1}) \leq 6H^2 d\sqrt{K \log(4H^3 K B^2/\delta) \log(1 + KH^2 B^2/d)}. \tag{55}$$

$$\square$$

**Lemma 9.** *Under event $\mathcal{E}_1$, $\tilde{V}_1^*(s_1, R_K) \leq 6H^2 d\sqrt{\frac{\log(4H^3 K B^2/\delta) \log(1 + KH^2 B^2/d)}{K}}$.*

*Proof.* Since $\tilde{V}_1^*(s_{k,1}, R_k) \leq V_{k,1}(s_{k,1})$ by lemma 7, we have

$$\sum_{k=1}^K \tilde{V}_1^*(s_1, R_k) \leq \sum_{k=1}^K V_{k,1}(s_1) \leq 6H^2 d\sqrt{K \log(4H^3KB^2/\delta)\log(1+KH^2B^2/d)}. \quad (56)$$

By the definition of $R_k$, we know that $R_k(s,a)$ is non-increasing w.r.t $k$. Therefore,

$$\sum_{k=1}^K \tilde{V}_1^*(s_1, R_K) \leq \sum_{k=1}^K \tilde{V}_1^*(s_1, R_k) \leq 6H^2 d\sqrt{K \log(4H^3KB^2/\delta)\log(1+KH^2B^2/d)}. \quad (57)$$

The lemma is proved by dividing both sides by $K$. $\qquad\square$

### A.7 PROOF OF THEOREM 1

Combining the results in Lemma 9 and Lemma 6, we know that for any reward function $R$,

$$V_1^*(s_1, R) - V_1^{\hat{\pi}_R}(s_1, R) \leq 24H^2 d\sqrt{\frac{\log(4H^3KB^2/\delta)\log(1+KH^2B^2/d)}{K}} + \epsilon_{\text{opt}}. \quad (58)$$

Choosing $K = \frac{C_1 H^4 d^2 \log(4H^3KB^2/\delta)\log(1+KH^2B^2/d)}{\epsilon^2}$ for some constant $C_1$ suffices to guarantee that $V_1^*(s_1, R) - V_1^{\hat{\pi}_R}(s_1, R) \leq \epsilon + \epsilon_{\text{opt}}$.

## B OMITTED DETAILS IN SECTION 5

### B.1 NOTATIONS

In this subsection, we summarize the notations used in Section B.

| Symbol | Explanation |
|---|---|
| $\mathcal{E}_2$ | The high-probability event for Theorem 2 |
| $s_{k,h}, a_{k,h}$ | The state and action that the agent encounters in episode $k$ and step $h$ |
| $\phi_V(s,a,R)$ | $\sum_{s'} \phi(s,a,s')V^{\pi,P}(s',R)$ for certain value function $V$ |
| $\hat{V}_h^{\pi,\tilde{P}}(s,R)$ | The value function of policy $\pi$ in the MDP model with transition $\tilde{P}$ and reward $R$ |
| $\tilde{V}_{k,h}^{\pi,\tilde{P}}(s,R)$ | The expected uncertainty along the trajectories induced by policy $\pi$ for $\hat{V}_h^{\pi,\tilde{P}}(s,R)$ (Defined in Eqn 5) |
| $V_{k,h}^{\pi,\tilde{P}}(s,R)$ | The optimistic estimation of $\tilde{V}_{k,h}^{\pi,\tilde{P}}(s,R)$ (Defined in Eqn 7) |
| $u_{1,k,h}^{\pi,\tilde{P}}(s,a,R)$ | The exploration-driven reward for transition $\tilde{P}$, reward $R$ and policy $\pi$ (Defined in Eqn 4) |
| $u_{2,k,h}^{\tilde{P},\pi}(s,a,R)$ | The confidence bonus ensuring the optimism $V_{k,h}^{\pi,\tilde{P}}(s,R) \geq \tilde{V}_{k,h}^{\pi,\tilde{P}}(s,R)$ (Defined in Eqn 6) |
| $\mathbb{V}_h V(s,a)$ | The one-step transition variance w.r.t. certain value function $V$ |
| $\bar{\mathbb{V}}_{1,k,h}(s,a)$ | The empirical variance estimation w.r.t. the value $\hat{V}_h^{\pi_k,\tilde{P}_k}(s,R_k)$ (Defined in Eqn 59) |
| $\bar{\mathbb{V}}_{2,k,h}(s,a)$ | The empirical variance estimation w.r.t. the value $V_h^{\pi_k,\tilde{P}_k}(s,R_k)$ (Defined in Eqn 60) |
| $E_{1,k,h}, E_{2,k,h}$ | The confidence bonus for the variance estimation $\bar{\mathbb{V}}_{1,k,h}(s,a)$ and $\bar{\mathbb{V}}_{2,k,h}(s,a)$, respectively |
| $\bar{\sigma}_{1,k,h}^2, \bar{\sigma}_{2,k,h}^2$ | The optimistic variance estimation for $\bar{\mathbb{V}}_{1,k,h}(s,a)$ and $\bar{\mathbb{V}}_{2,k,h}(s,a)$, respectively |
| $\hat{\theta}_{i,k,h}$ | The parameter estimation w.r.t. certain value function (Defined in Section B.2) |
| $\Lambda_{i,k,h}$ | The empirical covariance matrix w.r.t. certain value function (Defined in Section B.2) |
| $\mathcal{U}_{k,h}$ | The confidence set containing $\theta_h$ with high probability |
| $\pi_k, \tilde{\theta}_k, R_k$ | $\arg\max_{\pi,\tilde{\theta}_h \in \mathcal{U}_{k,h}, R} V_{k,1}^{\pi,\tilde{P}}(s_1, R)$ |
| $\tilde{Y}_{k,h}(s)$ | The "value function" for the MDP with transition $\tilde{P}_k$ and reward $\bar{\mathbb{V}}_{1,k,h}$ |
| $\tau$ | $\log(32K^2H/\delta)\log^2(1+KH^4B^2)$ |

### B.2 OMITTED DETAILS OF ALGORITHM 2

In algorithm 2, we maintain five different parameter estimation $\hat{\theta}_{1,k,h}, \hat{\theta}_{2,k,h}, \hat{\theta}_{3,k,h}, \hat{\theta}_{4,k,h}$ and $\hat{\theta}_{5,k,h}$, and the corresponding covariance matrix $\Lambda_{1,k,h}, \Lambda_{2,k,h}, \Lambda_{3,k,h}, \Lambda_{4,k,h}$ and $\Lambda_{5,k,h}$.

$\hat{\theta}_{1,k,h}, \hat{\theta}_{2,k,h}$ are the parameter estimation using history samples w.r.t the features of variance-normalized value function $\hat{V}_{k,h+1}^{\pi_k,\tilde{P}_k}/\bar{\sigma}_{1,k,h}$ and $V_{k,h+1}^{\pi_k,\tilde{P}_k}/\bar{\sigma}_{2,k,h}$, respectively. $\hat{\theta}_{3,k,h}, \hat{\theta}_{4,k,h}$ are the parameter estimation using samples w.r.t the features of value function $\left(\hat{V}_{k,h+1}^{\pi_k,\tilde{P}_k}\right)^2$ and $\left(V_{k,h+1}^{\pi_k,\tilde{P}_k}\right)^2$. $\hat{\theta}_{5,k,h}$ is somewhat technical and will be explained later. for any $i \in [5]$ and $h \in [H]$, $\hat{\theta}_{i,1,h}$ are initialized as $\mathbf{0}$, and $\Lambda_{i,1,h}$ are initialized as $\lambda I$ in Algorithm 2. $\mathcal{U}_{1,h}$ is the set containing all the $\tilde{\theta}_h$ that makes $\tilde{P}_h$ well-defined, i.e. $\sum_{s'} \tilde{\theta}_h^\top \phi(s'|s,a) = 1$ and $\tilde{\theta}_h^\top \phi(s'|s,a) \geq 0, \forall s,a$.

After observing $\{s_{k,h}, a_{k,h}\}_{h=1}^H$ in episode $k$, we calculate $\bar{\mathbb{V}}_{1,k,h}(s,a)$ and $\bar{\mathbb{V}}_{2,k,h}(s,a)$ as the corresponding variance in the empirical MDP with transition dynamics $\tilde{P}_k$.

$$\bar{\mathbb{V}}_{1,k,h}(s,a) = \tilde{\theta}_{k,h}^\top \sum_{s'} \phi(s,a,s') \left(\hat{V}_{k,h+1}^{\pi_k,\tilde{P}_k}(s',R_k)\right)^2 - \left[\tilde{\theta}_{k,h}^\top \sum_{s'} \phi(s,a,s')\hat{V}_{k,h+1}^{\pi_k,\tilde{P}_k}(s',R_k)\right]^2,$$
(59)

$$\bar{\mathbb{V}}_{2,k,h}(s,a) = \tilde{\theta}_{k,h}^\top \sum_{s'} \phi(s,a,s') \left(V_{k,h+1}^{\pi_k,\tilde{P}_k}(s',R_k)\right)^2 - \left[\tilde{\theta}_{k,h}^\top \sum_{s'} \phi(s,a,s')V_{k,h+1}^{\pi_k,\tilde{P}_k}(s',R_k)\right]^2.$$
(60)

To guarantee the variance estimation is optimistic, we calculate the confidence bonus $E_{1,k,h}$ and $E_{2,k,h}$ for variance estimation $\bar{\mathbb{V}}_{1,k,h}(s_{k,h}, a_{k,h})$ and $\bar{\mathbb{V}}_{2,k,h}(s_{k,h}, a_{k,h})$:

$$E_{1,k,h} = \min\left\{H^2, 4H\check{\beta}_k \left\|\sum_{s'} \phi(s_{k,h},a_{k,h},s')\hat{V}_{k,h+1}^{\pi_k,\tilde{P}_k}(s',R_k)\right\|_{(\Lambda_{1,k,h})^{-1}}\right\}$$
(61)

$$+ \min\left\{H^2, 2\tilde{\beta}_k \left\|\sum_{s'} \phi(s_{k,h},a_{k,h},s') \left(\hat{V}_{k,h+1}^{\pi_k,\tilde{P}_k}(s',R_k)\right)^2\right\|_{(\Lambda_{3,k,h})^{-1}}\right\},$$
(62)

$$E_{2,k,h} = \min\left\{H^2, 4H\check{\beta}_k \left\|\sum_{s'} \phi(s_{k,h},a_{k,h},s')V_{k,h+1}^{\pi_k,\tilde{P}_k}(s',R_k)\right\|_{(\Lambda_{2,k,h})^{-1}}\right\}$$
(63)

$$+ \min\left\{H^2, 2\tilde{\beta}_k \left\|\sum_{s'} \phi(s_{k,h},a_{k,h},s') \left(V_{k,h+1}^{\pi_k,\tilde{P}_k}(s',R_k)\right)^2\right\|_{(\Lambda_{4,k,h})^{-1}}\right\}.$$
(64)

We add the bonuses to $\bar{\mathbb{V}}_{1,k,h}(s_{k,h}, a_{k,h})$ and $\bar{\mathbb{V}}_{2,k,h}(s_{k,h}, a_{k,h})$ and maintain the optimistic variance estimation $\bar{\sigma}_{1,k,h}^2$ and $\bar{\sigma}_{2,k,h}^2$:

$$\bar{\sigma}_{1,k,h}^2 = \max\left\{H^2/d, \bar{\mathbb{V}}_{1,k,h}(s_{k,h}, a_{k,h}) + E_{1,k,h}\right\},$$
(65)

$$\bar{\sigma}_{2,k,h}^2 = \max\left\{H^2/d, \bar{\mathbb{V}}_{2,k,h}(s_{k,h}, a_{k,h}) + E_{2,k,h}\right\}.$$
(66)

We use $\bar{\sigma}_{i,k,h}$ to normalize the value obtained in each step. By setting $\lambda = B^{-2}$ and solving the ridge regression defined in Eqn. 8, we know that $\hat{\theta}_{i,k+1,h}$ and $\Lambda_{i,k+1,h}$ are updated in the following

way:

$$\Lambda_{1,k+1,h} = \Lambda_{1,k,h} + (\bar{\sigma}_{1,k,h})^{-2} \left( \sum_{s'} \phi(s_{k,h}, a_{k,h}, s') \hat{V}_{k,h+1}^{\pi_k, \tilde{P}_k}(s', R_k) \right) \left( \sum_{s'} \phi(s_{k,h}, a_{k,h}, s') \hat{V}_{k,h+1}^{\pi_k, \tilde{P}_k}(s', R_k) \right)^{\top},$$

(67)

$$\Lambda_{2,k+1,h} = \Lambda_{1,k,h} + (\bar{\sigma}_{2,k,h})^{-2} \left( \sum_{s'} \phi(s_{k,h}, a_{k,h}, s') V_{k,h+1}^{\pi_k, \tilde{P}_k}(s', R_k) \right) \left( \sum_{s'} \phi(s_{k,h}, a_{k,h}, s') V_{k,h+1}^{\pi_k, \tilde{P}_k}(s', R_k) \right)^{\top},$$

(68)

$$\hat{\theta}_{1,k+1,h} = (\Lambda_{1,k+1,h})^{-1} \sum_{t=1}^{k} (\bar{\sigma}_{1,t,h})^{-2} \left( \sum_{s'} \phi(s_{t,h}, a_{t,h}, s') \hat{V}_{t,h+1}^{\pi_t, \tilde{P}_t}(s', R_t) \right) \hat{V}_{t,h+1}^{\pi_t, \tilde{P}_t}(s_{t,h+1}, R_t),$$

(69)

$$\hat{\theta}_{2,k+1,h} = (\Lambda_{2,k+1,h})^{-1} \sum_{t=1}^{k} (\bar{\sigma}_{2,t,h})^{-2} \left( \sum_{s'} \phi(s_{t,h}, a_{t,h}, s') V_{t,h+1}^{\pi_t, \tilde{P}_t}(s', R_t) \right) V_{t,h+1}^{\pi_t, \tilde{P}_t}(s_{t,h+1}, R_t).$$

(70)

Similarly, we update the estimation w.r.t $\left( \hat{V}_{k,h+1}^{\pi_k, \tilde{P}_k}(s', R_k) \right)^2$ and $\left( V_{k,h+1}^{\pi_k, \tilde{P}_k}(s', R_k) \right)^2$:

$$\Lambda_{3,k+1,h} = \Lambda_{3,k,h} + \left( \sum_{s'} \phi(s_{k,h}, a_{k,h}, s') \left( \hat{V}_{k,h+1}^{\pi_k, \tilde{P}_k}(s', R_k) \right)^2 \right) \left( \sum_{s'} \phi(s_{k,h}, a_{k,h}, s') \left( \hat{V}_{k,h+1}^{\pi_k, \tilde{P}_k}(s', R_k) \right)^2 \right)^{\top},$$

(71)

$$\Lambda_{4,k+1,h} = \Lambda_{4,k,h} + \left( \sum_{s'} \phi(s_{k,h}, a_{k,h}, s') \left( V_{k,h+1}^{\pi_k, \tilde{P}_k}(s', R_k) \right)^2 \right) \left( \sum_{s'} \phi(s_{k,h}, a_{k,h}, s') \left( V_{k,h+1}^{\pi_k, \tilde{P}_k}(s', R_k) \right)^2 \right)^{\top},$$

(72)

$$\hat{\theta}_{3,k+1,h} = (\Lambda_{3,k+1,h})^{-1} \sum_{t=1}^{k} \left( \sum_{s'} \phi(s_{t,h}, a_{t,h}, s') \left( \hat{V}_{t,h+1}^{\pi_t, \tilde{P}_t}(s', R_t) \right)^2 \right) \left( \hat{V}_{t,h+1}^{\pi_t, \tilde{P}_t}(s_{t,h+1}, R_t) \right)^2,$$

(73)

$$\hat{\theta}_{4,k+1,h} = (\Lambda_{4,k+1,h})^{-1} \sum_{t=1}^{k} \left( \sum_{s'} \phi(s_{t,h}, a_{t,h}, s') \left( V_{t,h+1}^{\pi_t, \tilde{P}_t}(s', R_t) \right)^2 \right) \left( V_{t,h+1}^{\pi_t, \tilde{P}_t}(s_{t,h+1}, R_t) \right)^2.$$

(74)

We define $\tilde{Y}_{k,h}$ to be the "value function" with transition $\tilde{P}_k$ and reward $\bar{\mathbb{V}}_{1,k,h}$.

$$\tilde{Y}_{k,H+1}(s) = 0,$$

(75)

$$\tilde{Y}_{k,h}(s) = \bar{\mathbb{V}}_{1,k,h}(s, \pi_k(s)) + \tilde{P}_{k,h} \tilde{Y}_{k,h+1}(s, \pi_k(s)).$$

(76)

$\hat{\theta}_{5,k,h}$ is the parameter estimation using samples w.r.t. $\tilde{Y}_{k,h}(s)$:

$$\Lambda_{5,k+1,h} = \Lambda_{5,k,h} + \left( \sum_{s'} \phi(s_{k,h}, a_{k,h}, s') \tilde{Y}_{k,h+1}(s') \right) \left( \sum_{s'} \phi(s_{k,h}, a_{k,h}, s') \tilde{Y}_{k,h+1}(s') \right)^{\top},$$

(77)

$$\hat{\theta}_{5,k+1,h} = (\Lambda_{5,k+1,h})^{-1} \sum_{t=1}^{k} \left( \sum_{s'} \phi(s_{t,h}, a_{t,h}, s') \tilde{Y}_{t,h+1}(s') \right) \tilde{Y}_{t,h+1}(s_{k,h+1}).$$

(78)

We update the high-confidence set $\mathcal{U}_{k,h}$ by adding following five constraints:

$$\|\tilde{\theta}_h - \hat{\theta}_{1,k,h}\|_{\Lambda_{1,k,h}} \leq \hat{\beta}, \tag{79}$$

$$\|\tilde{\theta}_h - \hat{\theta}_{2,k,h}\|_{\Lambda_{2,k,h}} \leq \hat{\beta}, \tag{80}$$

$$\|\tilde{\theta}_h - \hat{\theta}_{3,k,h}\|_{\Lambda_{3,k,h}} \leq \tilde{\beta}, \tag{81}$$

$$\|\tilde{\theta}_h - \hat{\theta}_{4,k,h}\|_{\Lambda_{4,k,h}} \leq \tilde{\beta}, \tag{82}$$

$$\|\tilde{\theta}_h - \hat{\theta}_{5,k,h}\|_{\Lambda_{5,k,h}} \leq \tilde{\beta}. \tag{83}$$

Note that we add new constraints to this confidence set $\mathcal{U}_{k,h}$ in each episode, instead of updating $\mathcal{U}_{k,h}$ as the intersection of the new constraints. This operation is designed to ensure that the cardinality of the confidence set $\mathcal{U}_{k,h}$ is non-increasing w.r.t. the episode $k$, so that $V_{k,1}^{\pi,\tilde{P}}(s_1, R)$ is always non-increasing w.r.t. $k$.

During the proof, we use $\phi_{V^{\pi,P}}(s, a, R)$ as a shorthand of $\sum_{s'} \phi(s, a, s') V^{\pi,P}(s', R)$.

### B.3 High-probability events

For notation convenience, we use $\mathbb{V}_{1,k,h}(s, a)$ and $\mathbb{V}_{2,k,h}(s, a)$ to denote the one-step transition variance with regard to $\hat{V}_{k,h+1}^{\pi_k, P_k}$ and $V_{k,h+1}^{\pi_k, P_k}$, i.e.

$$\mathbb{V}_{1,k,h}(s, a) = \mathbb{E}_{s' \sim P_h(\cdot|s,a)} \left[ \left( \hat{V}_{k,h+1}^{\pi_k, P_k}(s', R_k) - P_h \hat{V}_{k,h}^{\pi_k, P_k}(s, a, R_k) \right)^2 \right], \tag{84}$$

$$\mathbb{V}_{2,k,h}(s, a) = \mathbb{E}_{s' \sim P_h(\cdot|s,a)} \left[ \left( V_{k,h+1}^{\pi_k, P_k}(s', R_k) - P_h V_{k,h}^{\pi_k, P_k}(s, a, R_k) \right)^2 \right]. \tag{85}$$

$$\tag{86}$$

**Lemma 10.** *With probability at least $1 - \delta$, the following event holds for any $h \in [H], k \in [K]$:*

$$\left\| \tilde{\theta}_{k,h} - \theta_h \right\|_{\Lambda_{1,k,h}} \leq 2\hat{\beta}, \tag{87}$$

$$\left\| \tilde{\theta}_{k,h} - \theta_h \right\|_{\Lambda_{2,k,h}} \leq 2\hat{\beta}, \tag{88}$$

$$\left| \bar{\mathbb{V}}_{1,k,h}(s_{k,h}, a_{k,h}) - \mathbb{V}_{1,k,h}(s_{k,h}, a_{k,h}) \right| \leq E_{1,k,h}, \tag{89}$$

$$\left| \bar{\mathbb{V}}_{2,k,h}(s_{k,h}, a_{k,h}) - \mathbb{V}_{2,k,h}(s_{k,h}, a_{k,h}) \right| \leq E_{2,k,h}, \tag{90}$$

$$\left\| \tilde{\theta}_{k,h} - \theta_h \right\|_{\Lambda_{5,k,h}} \leq 2\tilde{\beta}. \tag{91}$$

*Proof.* We firstly prove the first and the third inequality, the second and the fourth inequality can be proved following the same idea. By the definition of $\bar{\mathbb{V}}_{1,k,h}(s_{k,h}, a_{k,h})$ and $\mathbb{V}_{1,k,h}(s_{k,h}, a_{k,h})$:

$$\left| \bar{\mathbb{V}}_{1,k,h}(s_{k,h}, a_{k,h}) - \mathbb{V}_{1,k,h}(s_{k,h}, a_{k,h}) \right| \tag{92}$$

$$\leq \min \left\{ H^2, 2H \left\| \tilde{\theta}_{k,h} - \theta_h \right\|_{\Lambda_{1,k,h}} \left\| \phi_{\hat{V}_{k,h+1}^{\pi_k, P_k}}(s_{k,h}, a_{k,h}, R_k) \right\|_{(\Lambda_{1,k,h})^{-1}} \right\} \tag{93}$$

$$+ \min \left\{ H^2, \left\| \tilde{\theta}_{k,h} - \theta_h \right\|_{\Lambda_{3,k,h}} \left\| \phi_{\left( \hat{V}_{k,h+1}^{\pi_k, P_k} \right)^2}(s_{k,h}, a_{k,h}, R_k) \right\|_{(\Lambda_{3,k,h})^{-1}} \right\}. \tag{94}$$

Let $x_k = (\bar{\sigma}_{1,k,h})^{-1} \phi_{\hat{V}_{k,h+1}^{\pi_k, P_k}}(s_{k,h}, a_{k,h})$, and the noise

$$\eta_k = (\bar{\sigma}_{1,k,h})^{-1} \hat{V}_{k,h+1}^{\pi_k, P_k}(s_{k,h+1}, R_k) - (\bar{\sigma}_{1,k,h})^{-1} \left\langle \phi_{\hat{V}_{k,h+1}^{\pi_k, P_k}}(s_{k,h}, a_{k,h}, R_k), \theta_h \right\rangle.$$

Since $\bar{\sigma}_{1,k,h} \geq \sqrt{H^2/d}$, we have $\|x_k\|_2 \leq \sqrt{d}$, $\mathbb{E}\left[ \eta_k^2 \mid \mathcal{G}_k \right] \leq d$. By Lemma 22, we have with prob at least $1 - \delta/(8H)$, for all $k \leq K$,

$$\left\| \theta_h - \hat{\theta}_{1,k,h} \right\|_{\Lambda_{1,k,h}} \leq 16d\sqrt{\log(1 + KH^2/(d\lambda)) \log(32K^2 H/\delta)} + \sqrt{\lambda}B = \check{\beta}. \tag{95}$$

Similarly, we can prove that

$$\left\| \theta_h - \hat{\theta}_{3,k,h} \right\|_{\Lambda_{3,k,h}} \leq 16H^2 \sqrt{d \log(1 + KH^4/(d\lambda)) \log(32K^2H/\delta)} + \sqrt{\lambda}B = \check{\beta}. \quad (96)$$

Since $\left\| \tilde{\theta}_{k,h} - \hat{\theta}_{1,k,h} \right\|_{\Lambda_{1,k,h}} \leq \hat{\beta} \leq \check{\beta}$ by Inq 79, we have $\left\| \theta_h - \tilde{\theta}_{k,h} \right\|_{\Lambda_{1,k,h}} \leq 2\check{\beta}$. Similarly, since $\left\| \tilde{\theta}_{k,h} - \hat{\theta}_{3,k,h} \right\|_{\Lambda_{3,k,h}} \leq \tilde{\beta}$ by Inq 81, we have $\left\| \theta_h - \tilde{\theta}_{k,h} \right\|_{\Lambda_{3,k,h}} \leq 2\tilde{\beta}$. Plugging the above inequalities back to Inq. 92, we have

$$\left| \bar{\mathbb{V}}_{1,k,h}(s_{k,h}, a_{k,h}) - \mathbb{V}_{1,k,h}(s_{k,h}, a_{k,h}) \right| \leq E_{1,k,h}. \quad (97)$$

The above inequality indicates that $\bar{\sigma}_{1,k,h} \geq \mathbb{V}_{1,k,h}$, which is an optimistic estimation. As a result, the variance of the single-step noise $\eta_k = (\bar{\sigma}_{1,k,h})^{-1} \hat{V}_{k,h+1}^{\pi_k, P_k}(s_{k,h+1}, R_k) - (\bar{\sigma}_{1,k,h})^{-1} \left\langle \phi_{\hat{V}_{k,h+1}^{\pi_k, P_k}}(s_{k,h}, a_{k,h}, R_k), \theta_h \right\rangle$ satisfies $\mathbb{E}\left[\eta_k^2 \mid \mathcal{G}_k\right] \leq 1$. By Lemma 22, we can prove a tighter confidence guarantee for $\hat{\theta}_{1,k,h}$:

$$\left\| \theta_h - \hat{\theta}_{1,k,h} \right\|_{\Lambda_{1,k,h}} \leq 16 \sqrt{d \log(1 + KH^2/(d\lambda)) \log(32K^2H/\delta)} + \sqrt{\lambda}B = \hat{\beta}. \quad (98)$$

Combining with $\left\| \tilde{\theta}_{1,k,h} - \hat{\theta}_{1,k,h} \right\|_{\Lambda_{1,k,h}} \leq \hat{\beta}$ by Inq 79, we have $\left\| \theta_h - \tilde{\theta}_{1,k,h} \right\|_{\Lambda_{1,k,h}} \leq 2\hat{\beta}$.

Now we prove the last inequality in this lemma. Recall that we define

$$\tilde{Y}_{k,H+1}(s) = 0, \quad (99)$$
$$\tilde{Y}_{k,h}(s) = \bar{\mathbb{V}}_{1,k,h}(s, \pi_k(s)) + \tilde{P}_{k,h}\tilde{Y}_{k,h+1}(s, \pi_k(s)). \quad (100)$$

Note that $\tilde{Y}_{k,h}(s) \leq H^2$ by the law of total variance (Lattimore & Hutter, 2012; Azar et al., 2013). Therefore, the variance on the single-step noise in at most $H^2$. By Lemma 22, we have

$$\left\| \theta_h - \hat{\theta}_{5,k,h} \right\|_{\Lambda_{5,k,h}} \leq \tilde{\beta}. \quad (101)$$

Combining with $\left\| \tilde{\theta}_h - \hat{\theta}_{5,k,h} \right\|_{\Lambda_{5,k,h}} \leq \tilde{\beta}$ by Inq 83, we have $\left\| \theta_h - \tilde{\theta}_{5,k,h} \right\|_{\Lambda_{5,k,h}} \leq 2\tilde{\beta}$. $\qquad \square$

**Lemma 11.** *With probability at least $1 - \delta/2$, we have*

$$\sum_{k=1}^{K} \sum_{h=1}^{H} \left( P_h V_{k,h+1}^{\pi_k, \tilde{P}_k}(s_{k,h}, a_{k,h}, R_k) - V_{k,h+1}^{\pi_k, \tilde{P}_k}(s_{k,h+1}, a_{k,h+1}, R_k) \right) \leq \sqrt{2H^3 K \log(8/\delta)}, \quad (102)$$

$$\sum_{k=1}^{K} \sum_{h=1}^{H} \left( P_h \tilde{Y}_{k,h+1}(s_{k,h}, a_{k,h}, R_k) - \tilde{Y}_{k,h+1}(s_{k,h+1}, a_{k,h+1}, R_k) \right) \leq \sqrt{2H^5 K \log(8/\delta)}. \quad (103)$$

*Proof.* This lemma follows directly by Azuma's inequality for martingale difference sequence and union bound. $\qquad \square$

During the following analysis, we denote the high-probability events defined in Lemma 10 and Lemma 11 as $\mathcal{E}_2$.

### B.4 PROOF OF THEOREM 2

**Lemma 12.** *Under event $\mathcal{E}_2$, we have*

$$\sum_{k=1}^{K} V_{k,1}^{\pi_k, \tilde{P}_k}(s_1, R_k) \quad (104)$$

$$\leq O\left( \sqrt{(dH^4 + d^2H^3)K \log(KH/\delta) \log^2(KH^4B^2)} + (d^{2.5}H^2 + d^2H^3) \log(KH/\delta) \log^2(KH^4B^2) \right). \quad (105)$$

This lemma can be proved with the technique for regret analysis. We will explain it in Appendix B.5.

**Lemma 13.** *(Optimism) Under event $\mathcal{E}_2$, for any $h \in [H], k \in [K], s \in \mathcal{S}, \pi \in \Pi$, $R$ and $\tilde{P}$ satisfying $\tilde{P}_h = \tilde{\theta}_h^\top \phi$, we have*

$$V_{k,h}^{\pi,\tilde{P}}(s, R) \geq \tilde{V}_{k,h}^{\pi,\tilde{P}}(s, R). \tag{106}$$

*Proof.* This lemma is proved by induction. Suppose $V_{k,h+1}^{\pi,\tilde{P}}(s, R) \geq \tilde{V}_{k,h+1}^{\pi,\tilde{P}}(s, R)$, we have

$$V_{k,h}^{\pi,\tilde{P}}(s, R) - \tilde{V}_{k,h}^{\pi,\tilde{P}}(s, R) \tag{107}$$

$$\geq u_{2,k,h}^{\pi,\tilde{P}}(s, \pi(s), R) + (\tilde{P}_{k,h} - P_h)V_{k,h+1}^{\pi,\tilde{P}}(s, \pi(s), R) + P_h\left(V_{k,h+1}^{\pi,\tilde{P}} - \tilde{V}_{k,h+1}^{\pi,\tilde{P}}\right)(s, \pi(s), R) \tag{108}$$

$$\geq u_{2,k,h}^{\pi,\tilde{P}}(s, \pi(s), R) + (\tilde{P}_{k,h} - P_h)V_{k,h+1}^{\pi,\tilde{P}}(s, \pi(s), R) \tag{109}$$

$$\geq u_{2,k,h}^{\pi,\tilde{P}}(s, \pi(s), R) - \left\|\tilde{\theta}_{k,h} - \theta_h\right\|_{\Lambda_{2,k,h}} \left\|\sum_{s'} \phi(s, \pi(s), s')V_{k,h+1}^{\pi,\tilde{P}}(s', R)\right\|_{(\Lambda_{2,k,h})^{-1}} \tag{110}$$

$$\geq 0. \tag{111}$$

This indicates that $V_{k,h}^{\pi,\tilde{P}}(s, R) \geq \tilde{V}_{k,h}^{\pi,\tilde{P}}(s, R)$ holds for step $h$. $\qquad\square$

**Lemma 14.** *under event $\mathcal{E}_2$, we have $V_{K,1}^{\pi_K,\tilde{P}_K}(s_1, R_K) \leq \frac{\sum_{k=1}^K V_{k,1}^{\pi_k,\tilde{P}_k}(s_1, R_k)}{K}$.*

*Proof.* Firstly, we prove that $V_{k,1}^{\pi,\tilde{P}}(s, R)$ is non-increasing w.r.t $k$ for any fixed $\pi, \tilde{P}, s$ and $R$. This can be proved by induction. Suppose for any $k_1 \leq k_2$, $V_{k_1,h+1}^{\pi,\tilde{P}}(s, R) \geq V_{k_2,h+1}^{\pi,\tilde{P}}(s, R)$ for any $s$. Recall that

$$V_{k,h}^{\pi,\tilde{P}}(s, R) = \min\left\{u_{1,k,h}^{\pi,\tilde{P}}(s, \pi_h(s), R) + u_{2,k,h}^{\pi,\tilde{P}}(s, \pi_h(s), R) + \tilde{P}_h V_{k,h+1}^{\pi,\tilde{P}}(s, \pi_h(s), R), H\right\}. \tag{112}$$

Since $\Lambda_{1,k_1,h} \preccurlyeq \Lambda_{1,k_2,h}, \Lambda_{2,k_1,h} \preccurlyeq \Lambda_{2,k_2,h}$ and $V_{k_1,h+1}^{\pi,\tilde{P}}(s, R) \geq V_{k_2,h+1}^{\pi,\tilde{P}}(s, R)$ for any $s$, we can prove that

$$u_{1,k_1,h}^{\pi,\tilde{P}}(s, \pi_h(s), R) \geq u_{1,k_2,h}^{\pi,\tilde{P}}(s, \pi_h(s), R), \tag{113}$$

$$u_{2,k_1,h}^{\pi,\tilde{P}}(s, \pi_h(s), R) \geq u_{2,k_2,h}^{\pi,\tilde{P}}(s, \pi_h(s), R), \tag{114}$$

$$\tilde{P}_h V_{k_1,h+1}^{\pi,\tilde{P}}(s, \pi_h(s), R) \geq \tilde{P}_h V_{k_2,h+1}^{\pi,\tilde{P}}(s, \pi_h(s), R). \tag{115}$$

Therefore, $V_{k_1,h}^{\pi,\tilde{P}}(s, R) \geq V_{k_2,h}^{\pi,\tilde{P}}(s, R)$ holds for step $h$ and any $s$.

Since the cardinality of the transition set $\mathcal{U}_{k,h}$ is non-increasing (We add more constraints in each episode), we know that $\tilde{\theta}_{k_2,h} \in \mathcal{U}_{k_1,h}$. By the optimality of $\pi_{k_1}, \tilde{\theta}_{k_1}$ and $R_{k_1}$ in episode $k_1$, we have $V_{k_1,1}^{\pi_{k_1},\tilde{P}_{k_1}}(s_1, R_{k_1}) \geq V_{k_1,1}^{\pi_{k_2},\tilde{P}_{k_2}}(s_1, R_{k_2})$.

Combining the above two inequalities, we have

$$V_{k_1,1}^{\pi_{k_1},\tilde{P}_{k_1}}(s_1, R_{k,1}) \geq V_{k_1,1}^{\pi_{k_2},\tilde{P}_{k_2}}(s_1, R_{k_2}) \geq V_{k_2,1}^{\pi_{k_2},\tilde{P}_{k_2}}(s_1, R_{k_2}). \tag{116}$$

This indicates that the value $V_{k,1}^{\pi_k,\tilde{P}_k}(s_1, R_k)$ is non-increasing w.r.t. $k$. Therefore,

$$KV_{K,1}^{\pi_K,\tilde{P}_K}(s_1, R_K) \leq \sum_{k=1}^K V_{k,1}^{\pi_k,\tilde{P}_k}(s_1, R_k). \tag{117}$$

$\qquad\square$

**Lemma 15.** *Under event $\mathcal{E}_2$, the sub-optimality gap of the policy $\hat{\pi}_R$ returned in the planning phase for any reward function $R$ can be bounded by*

$$V_1^*(s_1, R) - V_1^{\hat{\pi}_R}(s_1, R) \le 4V_{K,1}^{\pi_K, \tilde{P}_K}(s_1, R_K) + \epsilon_{\text{opt}}. \tag{118}$$

*Proof.* Recall that we use $\hat{V}_h^\pi$ to denote the value function of policy $\pi$ on the estimated model $\tilde{\theta}_K$.

$$V_1^*(s_1, R) - V_1^{\hat{\pi}_R}(s_1, R) \tag{119}$$

$$= \left( V_1^*(s_1, R) - \hat{V}_1^{\pi_R^*, \tilde{P}_K}(s_1, R) \right) + \left( \hat{V}_1^{\hat{\pi}_R, \tilde{P}_K}(s_1, R) - V_1^{\hat{\pi}_R}(s_1) \right) + \left( \hat{V}_1^{\pi_R^*, \tilde{P}_K}(s_1, R) - \hat{V}_1^{\hat{\pi}_R, \tilde{P}_K}(s_1, R) \right) \tag{120}$$

$$\le V_1^*(s_1, R) - \hat{V}_1^{\pi_R^*, \tilde{P}_K}(s_1, R) + \hat{V}_1^{\hat{\pi}_R, \tilde{P}_K}(s_1, R) - V_1^{\hat{\pi}_R}(s_1, R) + \epsilon_{\text{opt}}. \tag{121}$$

The inequality is because that the policy $\hat{\pi}_R$ is the $\epsilon_{\text{opt}}$-optimal policy in the estimated MDP $\hat{M}$.

For notation convenience, we use $\text{traj} \sim (\pi, P)$ to denote that the trajectory $(\{s_h, a_h\}_{h=1}^H)$ is sampled with transition $P$ and policy $\pi$. For a certain sequence $\{R_h\}_{h=1}^H$, we define the function $W_h(\{R_h\})$ recursively from step $H + 1$ to step 1. Firstly, we define $W_{H+1}(\{R_h\}) = 0$. $W_h(R)$ is calculated from $W_{h+1}(R)$:

$$W_h(\{R_h\}) = \min\{H, R_h + W_{h+1}(\{R_h\})\}. \tag{122}$$

With the above notation, we can prove that for any policy $\pi \in \Pi$,

$$\left| \hat{V}_1^{\pi, \tilde{P}_K}(s_1, R) - V_1^\pi(s_1) \right| \tag{123}$$

$$\le \left| \mathbb{E}_{\text{traj} \sim (\pi, P)} W_1 \left( \left\{ (\tilde{P}_{K,h} - P_h) \hat{V}_{h+1}^{\pi, \tilde{P}_K}(s_h, a_h, R) \right\} \right) \right| \tag{124}$$

$$= \left| \mathbb{E}_{\text{traj} \sim (\pi, P)} W_1 \left( \left\{ (\tilde{\theta}_{K,h} - \theta_h) \sum_{s'} \phi(s_h, a_h, s') \hat{V}_{h+1}^{\pi, \tilde{P}_K}(s', R) \right\} \right) \right| \tag{125}$$

$$\le \mathbb{E}_{\text{traj} \sim (\pi, P)} W_1 \left( \left\{ \left\| \tilde{\theta}_{K,h} - \theta_h \right\|_{\Lambda_{1,K,h}} \left\| \sum_{s'} \phi(s_h, a_h, s') \hat{V}_{h+1}^{\pi, \tilde{P}_K}(s', R) \right\|_{(\Lambda_{1,K,h})^{-1}} \right\} \right) \tag{126}$$

$$\le \mathbb{E}_{\text{traj} \sim (\pi, P)} W_1 \left( \left\{ 2\hat{\beta} \left\| \sum_{s'} \phi(s_h, a_h, s') \hat{V}_{h+1}^{\pi, \tilde{P}_K}(s', R) \right\|_{(\Lambda_{1,K,h})^{-1}} \right\} \right) \tag{127}$$

$$= 2\mathbb{E}_{\text{traj} \sim (\pi, P)} W_1 \left( \left\{ u_{1,K,h}^{\pi, \tilde{P}_K}(s_h, a_h, R) \right\} \right) \tag{128}$$

$$= 2\tilde{V}_{K,1}^{\pi, \tilde{P}_K}(s_1, R) \tag{129}$$

$$\le 2V_{K,1}^{\pi, \tilde{P}_K}(s_1, R) \tag{130}$$

$$\le 2V_{K,1}^{\pi_K, \tilde{P}_K}(s_1, R_K). \tag{131}$$

The second inequality is due to lemma 10 . The third inequality is due to Lemma 13. The last inequality is due to the optimality of $\pi_K$ and $R_K$. Plugging this inequality back to Inq. 119, we can prove the lemma. $\square$

*Proof.* (Proof of Theorem 2) Combining Lemma 12, Lemma 14 and Lemma 15, we know that $K = \frac{C_2(d^2H^3 + dH^4) \log(dH/(\delta\epsilon)) \log^2(KHB/(\delta\epsilon))}{\epsilon^2} + \frac{C_2(d^{2.5}H^2 + d^2H^3) \log(dH/(\delta\epsilon)) \log^2(KHB/(\delta\epsilon))}{\epsilon}$ for some constant $C_2$ suffices to guarantee that $V_1^*(s_1, R) - V_1^{\hat{\pi}}(s_1, R) \le \epsilon + \epsilon_{\text{opt}}$. $\square$

## B.5 PROOF OF LEMMA 12

*Proof.* From the standard value decomposition technique (Azar et al., 2017; Jin et al., 2018), we can decompose $V_{k,1}^{\pi_k,\tilde{P}_k}(s_1, R_k)$ into following terms:

$$V_{k,1}^{\pi_k,\tilde{P}_k}(s_1, R_k) \tag{132}$$

$$\leq \sum_{h=1}^{H} \min\left\{H, u_{1,k,h}^{\pi_k,\tilde{P}_k}(s_{k,h}, a_{k,h}, R_k)\right\} + \sum_{h=1}^{H} \min\left\{H, u_{2,k,h}^{\pi_k,\tilde{P}_k}(s_{k,h}, a_{k,h}, R_k)\right\} \tag{133}$$

$$+ \sum_{h=1}^{H} \min\left\{H, (\tilde{P}_{k,h} - P_h)V_{k,h+1}^{\pi_k,\tilde{P}_k}(s_{k,h}, a_{k,h}, R_k)\right\} \tag{134}$$

$$+ \sum_{h=1}^{H} \left(P_h V_{k,h+1}^{\pi_k,\tilde{P}_k}(s_{k,h}, a_{k,h}, R_k) - V_{k,h+1}^{\pi_k,\tilde{P}_k}(s_{k,h+1}, a_{k,h+1}, R_k)\right) \tag{135}$$

$$\leq \sum_{h=1}^{H} \min\left\{H, u_{1,k,h}^{\pi_k,\tilde{P}_k}(s_{k,h}, a_{k,h}, R_k)\right\} \tag{136}$$

$$+ 2\sum_{h=1}^{H} \min\left\{H, u_{2,k,h}^{\pi_k,\tilde{P}_k}(s_{k,h}, a_{k,h}, R_k)\right\} \tag{137}$$

$$+ \sum_{h=1}^{H} \left(P_h V_{k,1}^{\pi_k,\tilde{P}_k}(s_{k,h}, a_{k,h}, R_k) - V_{k,1}^{\pi_k,\tilde{P}_k}(s_{k,h+1}, a_{k,h+1}, R_k)\right), \tag{138}$$

where the second inequality is derived from

$$(\tilde{P}_{k,h} - P_h)V_{k,h+1}^{\pi_k,\tilde{P}_k}(s_{k,h}, a_{k,h}, R_k) = (\tilde{\theta}_{k,h} - \theta_h)\sum_{s'} \phi(s_{k,h}, a_{k,h}, s')V_{k,h+1}^{\pi_k,\tilde{P}_k}(s', R_k) \tag{139}$$

$$\leq \left\|\tilde{\theta}_{k,h} - \theta_h\right\|_{\Lambda_{2,k,h}} \left\|\sum_{s'} \phi(s_{k,h}, a_{k,h}, s')V_{k,h+1}^{\pi_k,\tilde{P}_k}(s', R_k)\right\|_{\Lambda_{2,k,h}^{-1}} \tag{140}$$

$$\leq u_{2,k,h}^{\pi_k,\tilde{P}_k}(s_{k,h}, a_{k,h}, R_k). \tag{141}$$

Eqn 138 is a martingale difference sequence. By Lemma 11, the summation over all $k \in [K]$ is at most $\sqrt{2H^3 K \log(4/\delta)}$. We mainly focus on Eqn 136 and Eqn 137.

**Upper bound of Eqn 136** Firstly, we bound the summation of $\min\left\{H, u_{1,k,h}^{\pi_k,\tilde{P}_k}(s_{k,h}, a_{k,h}, R_k)\right\}$.

$$\sum_{k=1}^{K}\sum_{h=1}^{H}\min\left\{H, u_{1,k,h}^{\pi_k,\tilde{P}_k}(s_{k,h}, a_{k,h}, R_k)\right\} \tag{142}$$

$$=\sum_{k=1}^{K}\sum_{h=1}^{H}\min\left\{H, \hat{\beta}\left\|\sum_{s'}\phi(s_{k,h}, a_{k,h}, s')\hat{V}_{k,h+1}^{\pi_k,\tilde{P}_k}(s', R_k)\right\|_{\Lambda_{1,k,h}^{-1}}\right\} \tag{143}$$

$$\le\sum_{k=1}^{K}\sum_{h=1}^{H}\hat{\beta}\bar{\sigma}_{1,k,h}\min\left\{1, \left\|\sum_{s'}\phi(s_{k,h}, a_{k,h}, s')\frac{\hat{V}_{k,h+1}^{\pi_k,\tilde{P}_k}(s', R_k)}{\bar{\sigma}_{1,k,h}}\right\|_{\Lambda_{1,k,h}^{-1}}\right\} \tag{144}$$

$$\le\hat{\beta}\sqrt{\sum_{k,h}\bar{\sigma}_{1,k,h}^2\sum_{k,h}\min\left\{1, \left\|\sum_{s'}\phi(s_{k,h}, a_{k,h}, s')\frac{\hat{V}_{k,h+1}^{\pi_k,\tilde{P}_k}(s', R_k)}{\bar{\sigma}_{1,k,h}}\right\|_{\Lambda_{1,k,h}^{-1}}^2\right\}} \tag{145}$$

$$\le\hat{\beta}\sqrt{\sum_{k,h}\bar{\sigma}_{1,k,h}^2\cdot 2dH\log(1 + KH/\lambda)}. \tag{146}$$

The first inequality is due to Cauchy-Schwarz inequality. The second inequality is due to Lemma 21. By the definition of $\bar{\sigma}_{1,k,h}^2$, we have

$$\bar{\sigma}_{1,k,h}^2 \le H^2/d + E_{1,k,h} + \bar{\mathbb{V}}_{1,k,h}(s_{k,h}, a_{k,h}). \tag{147}$$

Now we bound $\sum_{k,h}E_{1,k,h}$ and $\sum_{k,h}\bar{\mathbb{V}}_{1,k,h}(s_{k,h}, a_{k,h})$ respectively. For $\sum_{k,h}E_{1,k,h}$, we have

$$\sum_{k,h}E_{1,k,h}\le\sum_{h=1}^{H}\sum_{k=1}^{K}\min\left\{H^2, 4H\check{\beta}\left\|\sum_{s'}\phi(s_{k,h}, a_{k,h}, s')\hat{V}_{k,h+1}^{\pi_k,\tilde{P}_k}(s', R_k)\right\|_{(\Lambda_{1,k,h})^{-1}}\right\} \tag{148}$$

$$+\sum_{h=1}^{H}\sum_{k=1}^{K}\min\left\{H^2, 2\tilde{\beta}\left\|\sum_{s'}\phi(s_{k,h}, a_{k,h}, s')\left(\hat{V}_{k,h+1}^{\pi_k,\tilde{P}_k}(s', R_k)\right)^2\right\|_{(\Lambda_{3,k,h})^{-1}}\right\}. \tag{149}$$

For the first part,

$$\sum_{h=1}^{H}\sum_{k=1}^{K}\min\left\{H^2, 4H\check{\beta}\left\|\sum_{s'}\phi(s_{k,h}, a_{k,h}, s')\hat{V}_{k,h+1}^{\pi_k,\tilde{P}_k}(s', R_k)\right\|_{(\Lambda_{1,k,h})^{-1}}\right\} \tag{150}$$

$$\le 4H\sum_{h=1}^{H}\sum_{k=1}^{K}\check{\beta}\bar{\sigma}_{1,k,h}\min\left\{1, \left\|\sum_{s'}\phi(s_{k,h}, a_{k,h}, s')\hat{V}_{k,h+1}^{\pi_k,\tilde{P}_k}(s', R_k)/\bar{\sigma}_{1,k,h}\right\|_{(\Lambda_{1,k,h})^{-1}}\right\} \tag{151}$$

$$\le 4H\check{\beta}\sqrt{\sum_{k,h}\bar{\sigma}_{1,k,h}^2\sum_{k,h}\min\left\{1, \left\|\sum_{s'}\phi(s_{k,h}, a_{k,h}, s')\frac{\hat{V}_{k,h+1}^{\pi_k,\tilde{P}_k}(s', R_k)}{\bar{\sigma}_{1,k,h}}\right\|_{\Lambda_{1,k,h}^{-1}}^2\right\}} \tag{152}$$

$$\le 4H\check{\beta}\sqrt{\sum_{k,h}\bar{\sigma}_{1,k,h}^2\cdot 2dH\log(1 + KH/\lambda)}, \tag{153}$$

where the last inequality is due to Lemma 21. Similarly, for the second part, by Lemma 21, we have

$$\sum_{h=1}^{H}\sum_{k=1}^{K}\min\left\{H^2, 2\tilde{\beta}\left\|\sum_{s'}\phi(s_{k,h}, a_{k,h}, s')\left(\hat{V}_{k,h+1}^{\pi_k,\tilde{P}_k}(s', R_k)\right)^2\right\|_{(\Lambda_{3,k,h})^{-1}}\right\} \tag{154}$$

$$\le 2\tilde{\beta}\sqrt{2dH^2K\log(1 + KH/\lambda)}. \tag{155}$$

Note that $\bar{\mathbb{V}}_{1,k,h}(s_{k,h}, a_{k,h})$ is the empirical variance of $\hat{V}_{k,h+1}^{\pi_k, \tilde{P}_k}(s', R_k)$ with transition $\tilde{P}_k(s'|s_{k,h}, a_{k,h})$. We bound the summation of $\bar{\mathbb{V}}_{1,k,h}(s_{k,h}, a_{k,h})$ by the law of total variance (Lattimore & Hutter, 2012; Azar et al., 2013).

Recall that we define

$$\tilde{Y}_{k,H+1}(s) = 0, \tag{156}$$

$$\tilde{Y}_{k,h}(s) = \bar{\mathbb{V}}_{1,k,h}(s, \pi_k(s)) + \tilde{P}_{k,h}\tilde{Y}_{k,h+1}(s, \pi_k(s)). \tag{157}$$

By the law of total variance, we have $\tilde{Y}_{k,1}(s) \leq H^2$ holds for any $s, a$ and $k$. We now bound the difference between $\sum_{k=1}^K \tilde{Y}_{k,1}(s_{k,1})$ and $\sum_{k=1}^K \sum_{h=1}^H \bar{\mathbb{V}}_{1,k,h}(s_{k,h}, a_{k,h})$.

$$\tilde{Y}_{k,1}(s_{k,1}) - \sum_{h=1}^H \bar{\mathbb{V}}_{1,k,h}(s_{k,h}, a_{k,h}) \tag{158}$$

$$= \tilde{P}_{k,1}\tilde{Y}_{k,2}(s_{k,1}, a_{k,1}) - \sum_{h=2}^H \bar{\mathbb{V}}_{1,k,h}(s_{k,h}, a_{k,h}) \tag{159}$$

$$\leq \min\left\{H^2, \left(\tilde{P}_{k,1} - P_1\right)\tilde{Y}_{k,2}(s_{k,1}, a_{k,1})\right\} + \left(P_1\tilde{Y}_{k,2}(s_{k,1}, a_{k,1}) - \tilde{Y}_{k,2}(s_{k,2})\right) \tag{160}$$

$$+ \left(\tilde{Y}_{k,2}(s_{k,2}) - \sum_{h=2}^H \bar{\mathbb{V}}_{1,k,h}(s_{k,h}, a_{k,h})\right). \tag{161}$$

Therefore, we have

$$\sum_{k=1}^K \tilde{Y}_{k,1}(s_{k,1}) - \sum_{k=1}^K \sum_{h=1}^H \bar{\mathbb{V}}_{1,k,h}(s_{k,h}, a_{k,h}) \tag{162}$$

$$\leq \sum_{k=1}^K \sum_{h=1}^H \min\left\{H^2, \left(\tilde{P}_{k,h} - P_h\right)\tilde{Y}_{k,h+1}(s_{k,h}, a_{k,h})\right\} \tag{163}$$

$$+ \sum_{h=1}^H \sum_{k=1}^K \left(P_h\tilde{Y}_{k,h+1}(s_{k,h}, a_{k,h}) - \tilde{Y}_{k,h+1}(s_{k,h+1})\right). \tag{164}$$

For Eqn 164, this term can be regarded as a martingale difference sequence, thus can be bounded by $H^2\sqrt{KH}$ by Lemma 11. For Eqn 163, we can bound this term in the following way:

$$\sum_{k=1}^K \sum_{h=1}^H \min\left\{H^2, \left(\tilde{P}_{k,h} - P_h\right)\tilde{Y}_{k,h+1}(s_{k,h}, a_{k,h})\right\} \tag{165}$$

$$\leq \sum_{k=1}^K \sum_{h=1}^H \min\left\{H^2, \left\|\tilde{\theta}_{k,h} - \theta_h\right\|_{\Lambda_{5,k,h}} \left\|\phi_{\tilde{Y}}(s_{k,h}, a_{k,h}, R)\right\|_{\Lambda_{5,k,h}^{-1}}\right\} \tag{166}$$

$$\leq \tilde{\beta} \sum_{k=1}^K \sum_{h=1}^H \min\left\{1, \left\|\phi_{\tilde{Y}}(s_{k,h}, a_{k,h}, R)\right\|_{\Lambda_{5,k,h}^{-1}}\right\} \tag{167}$$

$$\leq \tilde{\beta} \sum_{h=1}^H \sqrt{K \sum_{k=1}^K \min\left\{1, \left\|\phi_{\tilde{Y}}(s_{k,h}, a_{k,h}, R)\right\|_{\Lambda_{5,k,h}^{-1}}^2\right\}} \tag{168}$$

$$\leq \tilde{\beta}\sqrt{2dH^2K \log(1 + KH/\lambda)}. \tag{169}$$

The second inequality is due to Lemma 10. The third inequality is due to Cauchy-Schwarz inequality. The last inequality is due to Lemma 21.

From the above analysis, we have

$$\sum_{k,h} \bar{\mathbb{V}}_{1,k,h}(s_{k,h}, a_{k,h}) \leq H^2K + \tilde{\beta}\sqrt{2dH^2K \log(1 + KH/\lambda)}. \tag{170}$$

Therefore, the summation of $\bar{\sigma}_{1,k,h}^2$ can be bounded as

$$\sum_{k,h} \bar{\sigma}_{1,k,h}^2 \tag{171}$$

$$\leq H^3 K/d + \sum_{k,h} E_{1,k,h} + \sum_{k,h} \bar{\mathbb{V}}_{1,k,h} \tag{172}$$

$$\leq H^3 K/d + 4H\check{\beta}\sqrt{\sum_{k,h} \bar{\sigma}_{1,k,h}^2 \cdot 2dH \log(1 + KH/\lambda)} + 3\tilde{\beta}\sqrt{dH^2 K \log(1 + KH/\lambda)} + H^2 K. \tag{173}$$

We define $\tau = \log(32K^2 H/\delta) \log^2(1 + KH^4 B^2)$. Solving for $\sum_{k,h} \bar{\sigma}_{1,k,h}^2$, we have

$$\sum_{k,h} \bar{\sigma}_{1,k,h}^2 \leq c_1 \left( H^3 K/d + H^2 K + H^3 d^3 \tau + \sqrt{H^2 d^3 K \tau} \right) \tag{174}$$

$$\leq 2c_1 \left( H^3 K/d + H^2 K + H^3 d^3 \tau \right), \tag{175}$$

where $c_1$ denote a certain constant. The last inequality is due to $2\sqrt{ab} \leq a + b$ for $a, b \geq 0$. Plugging the above inequality back to Inq 142, we have for a constant $c_2$,

$$\sum_{k=1}^{K} \sum_{h=1}^{H} u_{1,k,h}^{\pi_k, \tilde{P}_k}(s_{k,h}, a_{k,h}, R_k) \leq c_2 \sqrt{(dH^4 + d^2 H^3) K\tau} + c_2 d^{2.5} H^2 \tau. \tag{176}$$

**Upper bound of Eqn 137**  Now we focus on the summation of $u_{2,k,h}^{\pi_k, \tilde{P}_k}(s_{k,h}, a_{k,h}, R_k)$. The proof follows almost the same arguments as the summation of $u_{1,k,h}^{\pi_k, \tilde{P}_k}(s_{k,h}, a_{k,h}, R_k)$, though the upper bound of $\sum_{k=1}^{K} \sum_{h=1}^{H} \bar{\mathbb{V}}_{2,k,h}(s_{k,h}, a_{k,h})$ is derived in a different way. Following the same proof idea, we can show that

$$\sum_{k=1}^{K} \sum_{h=1}^{H} u_{2,k,h}^{\pi_k, \tilde{P}_k}(s_{k,h}, a_{k,h}, R_k) \leq \hat{\beta}\sqrt{\sum_{k,h} \bar{\sigma}_{2,k,h}^2 2dH \log(1 + KH/\lambda)}, \tag{177}$$

$$\bar{\sigma}_{2,k,h}^2 \leq H^2/d + E_{2,k,h} + \bar{\mathbb{V}}_{2,k,h}(s_{k,h}, a_{k,h}), \tag{178}$$

$$\sum_{k=1}^{K} \sum_{h=1}^{H} E_{2,k,h} \leq 4H\check{\beta}\sqrt{\sum_{k,h} \bar{\sigma}_{2,k,h}^2 \cdot 2dH \log(1 + KH/\lambda)} + 2\tilde{\beta}\sqrt{2dH^2 K \log(1 + KH/\lambda)}. \tag{179}$$

Combining Inq 178 and Inq 179 and solving for $\sum_{k,h} \bar{\sigma}_{2,k,h}^2$, we have

$$\sum_{k,h} \bar{\sigma}_{2,k,h}^2 \tag{180}$$

$$\leq 2H^3 K/d + 16dH^3 \check{\beta}^2 \log(1 + KH/\lambda) + 4\tilde{\beta}\sqrt{dH^2 K \log(1 + KH/\lambda)} + 2\sum_{k,h} \bar{\mathbb{V}}_{2,k,h}(s_{k,h}, a_{k,h}) \tag{181}$$

$$\leq 6H^3 K/d + 16dH^3 \check{\beta}^2 \log(1 + KH/\lambda) + 4\tilde{\beta}^2 d^2/H \log(1 + KH/\lambda) + 2\sum_{k,h} \bar{\mathbb{V}}_{2,k,h}(s_{k,h}, a_{k,h}) \tag{182}$$

Plugging this inequality back to Inq 177, we have

$$\sum_{k=1}^{K} \sum_{h=1}^{H} u_{2,k,h}^{\pi_k, \tilde{P}_k}(s_{k,h}, a_{k,h}, R_k) \tag{183}$$

$$\leq \hat{\beta}\sqrt{6H^4 K \log(1 + KH/\lambda)} + 4\check{\beta}\hat{\beta}dH^2 \log(1 + KH/\lambda) + 2\tilde{\beta}\hat{\beta}d^{3/2} \log(1 + KH/\lambda) \tag{184}$$

$$+ \sqrt{32d^2 H \log(32K^2 H/\delta) \log^2(1 + KH^4 B^2) \sum_{k,h} \bar{\mathbb{V}}_{2,k,h}(s_{k,h}, a_{k,h})}. \tag{185}$$

We now bound the last term in the above equation. Define $\tau = \log(32K^2 H/\delta) \log^2(1 + KH^4 B^2)$, we have:

$$\sqrt{32 d^2 H \tau \sum_{k,h} \bar{\mathbb{V}}_{2,k,h}(s_{k,h}, a_{k,h})} \tag{186}$$

$$\leq \sqrt{32 d^2 H \tau \sum_{k=1}^{K} \sum_{h=1}^{H} \mathbb{E}_{s' \sim \tilde{P}_{k,h}(\cdot|s_{k,h}, a_{k,h})} \left[ (V_{k,h+1}^{\pi_k, \tilde{P}_k})^2(s', R_k) \right]} \tag{187}$$

$$\leq \sqrt{32 d^2 H^2 \tau \sum_{k=1}^{K} \sum_{h=1}^{H} \mathbb{E}_{s' \sim \tilde{P}_{k,h}(\cdot|s_{k,h}, a_{k,h})} \left[ V_{k,h+1}^{\pi_k, \tilde{P}_k}(s', R_k) \right]} \tag{188}$$

$$\leq 4\sqrt{2}\tilde{c}_3 d^2 H^3 \tau + \frac{1}{\tilde{c}_3 H} \sum_{k=1}^{K} \sum_{h=1}^{H} \mathbb{E}_{s' \sim \tilde{P}_{k,h}(\cdot|s_{k,h}, a_{k,h})} \left[ V_{k,h+1}^{\pi_k, \tilde{P}_k}(s', R_k) \right] \tag{189}$$

$$= 4\sqrt{2}\tilde{c}_3 d^2 H^3 \tau + \frac{1}{\tilde{c}_3 H} \sum_{k=1}^{K} \sum_{h=1}^{H} V_{k,h+1}^{\pi_k, \tilde{P}_k}(s_{k,h+1}, R_k) \tag{190}$$

$$+ \frac{1}{\tilde{c}_3 H} \sum_{k=1}^{K} \sum_{h=1}^{H} \left( P_h V_{k,h+1}^{\pi_k, \tilde{P}_k}(s_{k,h}, a_{k,h}, R_k) - V_{k,h+1}^{\pi_k, \tilde{P}_k}(s_{k,h+1}, R_k) \right) \tag{191}$$

$$+ \frac{1}{\tilde{c}_3 H} \sum_{k=1}^{K} \sum_{h=1}^{H} \left( \tilde{P}_{k,h} - P_h \right) V_{k,h+1}^{\pi_k, \tilde{P}_k}(s_{k,h}, a_{k,h}, R_k), \tag{192}$$

where $\tilde{c}_3 \geq 1$ is a constant to be defined later. The last inequality is due to $2\sqrt{ab} \leq a + b$ for any $a, b \geq 0$. Note that by Lemma 10, Eqn 192 is upper bounded by $\frac{1}{c_3 H} \sum_{k,h} u_{2,k,h}^{\pi_k, \tilde{P}_k}(s_{k,h}, a_{k,h}, R_k)$. Eqn 191 is a martingale difference sequence, and can be bounded by $\sqrt{HK \log(1/\delta)}$ with high probability. Plugging the above inequality back to Inq 183 and solving for $\sum_{k=1}^{K} \sum_{h=1}^{H} u_{2,k,h}^{\pi_k, \tilde{P}_k}(s_{k,h}, a_{k,h}, R_k)$, we have

$$\sum_{k=1}^{K} \sum_{h=1}^{H} u_{2,k,h}^{\pi_k, \tilde{P}_k}(s_{k,h}, a_{k,h}, R_k) \tag{193}$$

$$\leq c_3 \left( \sqrt{dH^4 K \tau} + (d^{2.5} H^2 + d^2 H^3)\tau + \frac{1}{\tilde{c}_3 H} \sum_{k=1}^{K} \sum_{h=1}^{H} V_{k,h+1}^{\pi_k, \tilde{P}_k}(s_{k,h+1}, R_k) \right), \tag{194}$$

where $c_3 \geq 1$ is a constant here. We set $\tilde{c}_3 = c_3$, then we have

$$\sum_{k=1}^{K} \sum_{h=1}^{H} u_{2,k,h}^{\pi_k, \tilde{P}_k}(s_{k,h}, a_{k,h}, R_k) \tag{195}$$

$$\leq c_3 \left( \sqrt{dH^4 K \tau} + (d^{2.5} H^2 + d^2 H^3)\tau \right) + \frac{1}{H} \sum_{k=1}^{K} \sum_{h=1}^{H} V_{k,h+1}^{\pi_k, \tilde{P}_k}(s_{k,h+1}, R_k), \tag{196}$$

**Bounding the summation of $V_{k,1}^{\pi_k, \tilde{P}_k}(s_1, R_k)$** In the above analysis, we derive the upper bound of Eqn 136 and Eqn 137 (Inq 176 and Inq 195). Now we can finally bound $\sum_{k=1}^{K} V_{k,1}^{\pi_k, \tilde{P}_k}(s_{k,1}, R_k)$. For a constant $c_4$,

$$\sum_{k=1}^{K} V_{k,1}^{\pi_k, \tilde{P}_k}(s_{k,1}, R_k) \tag{197}$$

$$\leq c_4 \sqrt{(dH^4 + d^2 H^3)K\tau} + c_4 (d^{2.5} H^2 + d^2 H^3)\tau + \frac{1}{H} \sum_{k=1}^{K} \sum_{h=1}^{H} V_{k,h+1}^{\pi_k, \tilde{P}_k}(s_{k,h+1}, R_k). \tag{198}$$

Following the same analysis, the above inequality actually also holds for any step $h \in [H]$:

$$\sum_{k=1}^{K} V_{k,h}^{\pi_k, \tilde{P}_k} (s_{k,h}, R_k) \tag{199}$$

$$\leq c_4 \sqrt{(dH^4 + d^2 H^3) K \tau} + c_4 (d^{2.5} H^2 + d^2 H^3) \tau + \frac{1}{H} \sum_{k=1}^{K} \sum_{h_1=h}^{H} V_{k,h_1+1}^{\pi_k, \tilde{P}_k}(s_{k,h_1+1}, R_k). \tag{200}$$

Define $G = c_4 \left( \sqrt{(dH^4 + d^2 H^3) K \tau} \right) + c_4 (d^{2.5} H^2 + d^2 H^3) \tau \right)$ and $a_h = \sum_{k=1}^{K} V_{k,h}^{\pi_k, \tilde{P}_k} (s_{k,h}, R_k)$. The above inequality can be simplified into the following form:

$$a_h \leq G + \frac{1}{H} \sum_{h_1=h}^{H} a_{h_1}, \tag{201}$$

and we have $a_{H+1} = 0$. From the elementary calculation, we can prove that $a_1 \leq (1 + \frac{1}{H})^H G \leq eG$. Therefore, we have

$$\sum_{k=1}^{K} V_{k,1}^{\pi_k, \tilde{P}_k} (s_{k,1}, R_k) \tag{202}$$

$$\leq c_5 \left( \sqrt{(dH^4 + d^2 H^3) K \log(KH/\delta) \log^2 (KH^4 B^2))} + (d^{2.5} H^2 + d^2 H^3) \log(KH/\delta) \log^2 (KH^4 B^2) \right). \tag{203}$$

for a constant $c_5 = e c_4$. $\qquad \square$

## C   THE LOWER BOUND FOR REWARD-FREE EXPLORATION

In this section, we prove that even in the setting of non-plug-in reward-free exploration, the sample complexity in the exploration phase to obtain an $\epsilon$-optimal policy is at least $\Omega(\frac{d^2 H^3}{\epsilon^2})$. We say an algorithm can $(\epsilon, \delta)$-learns the linear mixture MDP $\mathcal{M}$ if this algorithm returns an $\epsilon$-optimal policy for $\mathcal{M}$ with probability at least $1 - \delta$ in the planning phase after receiving samples for $K$ episodes in the exploration phase. Our theorem is stated as follows.

**Theorem 3.** *Suppose $\epsilon \leq \min \left( C_1 \sqrt{H}, C_2 \sqrt{dH^4 B} \right), B > 1, H > 4, d > 3$ for certain positive constants $C_1$ and $C_2$. Then for any algorithm $\mathcal{ALG}$ there exists an linear mixture MDP instance $\mathcal{M} = (\mathcal{S}, \mathcal{A}, P, R, H, \nu)$ such that if $\mathcal{ALG}$ $(\epsilon, \delta)$-learns the problem $\mathcal{M}$, $\mathcal{ALG}$ needs to collect at least $K = Cd^2 H^3 / \epsilon^2$ episodes during the exploration phase, where $C$ is an absolute constant, and $0 < \delta < 1$ is a positive constant that has no dependence on $\epsilon, H, d, K$.*

Compared with the lower bound proposed by Zhang et al. (2021a), we further improve their result by a factor of $d$. This lower bound also indicates that our sample complexity bound in Theorem 2 is statistically optimal. We prove Theorem 3 in Appendix C.1.

### C.1   PROOF OF THEOREM 3

Our basic idea to prove Theorem 3 is to connect the sample complexity lower bound with the regret lower bound of another constructed learning algorithm in the standard online exploration setting.

To start with, we notice that the reward-free exploration is strictly harder than the standard non-reward-free online exploration setting (e.g. Jin et al. (2018); Zhou et al. (2021; 2020a)), where the reward function is deterministic and known during the exploration. Therefore, if we can prove the sample complexity lower bound in the standard online exploration setting, this bound can also be applied to the reward-free setting.

For readers who are not familiar with the formulation of online exploration setting studied in this section, we firstly introduce the preliminaries. Compared with the reward-free exploration, the only

difference is that the reward $R(s, a)$ is fixed and known to the agent. In each episode, the agent starts from an initial state $s_{k,1}$ sampled from the distribution $\nu$. At each step $h \in [H]$, the agent observes the current state $s_{k,h} \in \mathcal{S}$, takes action $a_{k,h} \in \mathcal{A}$, receives the deterministic reward $R_h(s_{k,h}, a_{k,h})$, and transits to state $s_{k,h+1}$ with probability $P_h(s_{k,h+1}|s_{k,h}, a_{k,h})$. The episode ends when $s_{H+1}$ is reached. The agent's goal is to find a $\epsilon$-optimal policy $\pi$ after $K$ episodes. We say a policy $\pi$ is $\epsilon$-optimal if

$$\mathbb{E}\left[\sum_{h=1}^{H} R_h\left(s_h, a_h\right) \mid \pi\right] \geq \mathbb{E}\left[\sum_{h=1}^{H} R_h\left(s_h, a_h\right) \mid \pi^*\right] - \epsilon,$$

where $\pi^*$ is the optimal policy for the MDP $(\mathcal{S}, \mathcal{A}, P, R, H, \nu)$.

**Theorem 4.** *Suppose* $\epsilon \leq \min\left(C_1\sqrt{H}, C_2\sqrt{dH^4B}\right), B > 1, H > 4, d > 3$. *Then for any algorithm* $\mathcal{ALG}_1$ *solving the non-reward-free online exploration problem, there exists an linear mixture MDP* $\mathcal{M} = (\mathcal{S}, \mathcal{A}, P, R, H, \nu)$ *such that* $\mathcal{ALG}_1$ *needs to collect at least* $K = Cd^2H^3/\epsilon^2$ *episodes to output an $\epsilon$-optimal policy for the linear mixture MDP $\mathcal{M}$ with probability at least $1 - \delta$, where $C$ is an absolute constant, and $0 < \delta < 1$ is a positive constant that has no dependence on $\epsilon, H, d, K$.*

From the above discussion, Theorem 3 can be directly proved by reduction if Theorem 4 is true.

*Proof.* (Proof of Theorem 3) The theorem can be proved by contradiction. Suppose there is an algorithm $\mathcal{ALG}$ which can $(\epsilon, \delta)$-learns any linear mixture MDP instance $\mathcal{M}$ with only $K' \leq Cd^2H^3/\epsilon^2$ episodes. Then we can use this algorithm to solve the online exploration problem by simply ignoring the information about the reward function and directly calling the exploration algorithm of $\mathcal{ALG}$ during the exploration phase. Then in the planning phase, we use the planning algorithm of $\mathcal{ALG}$ to output a policy based on the reward function as well as samples collected in the exploration phase. Therefore, this indicates that $\mathcal{ALG}$ can output an $\epsilon$-optimal policy for the non-reward-free online exploration problem with probability at least $1 - \delta$ after only $K' \leq Cd^2H^3/\epsilon^2$ episodes. This contradicts the sample complexity lower bound in Theorem 4. $\qquad\square$

Now we discuss on how to prove Theorem 4.

*Proof.* (Proof of Theorem 4) Set $K_1 = cK$ for a positive constant $c \geq 2$. We construct another algorithm $\mathcal{ALG}_2$ for any possible $\mathcal{ALG}_1$ in the following way: In $\mathcal{ALG}_2$, the agent firstly runs $\mathcal{ALG}_1$ for $K_1/c = K$ episodes. After $K$ episodes, suppose $\mathcal{ALG}_1$ outputs a policy $\hat{\pi}$ according to certain policy distribution $\nu_\pi$. $\mathcal{ALG}_2$ executes the policy $\hat{\pi}$ in the following $\frac{c-1}{c}K_1$ episodes. The interaction ends after $K_1$ episodes. $\mathcal{ALG}_2$ can be regarded as an algorithm which firstly runs the online exploration algorithm $\mathcal{ALG}_1$ for $K$ episodes, and then evaluates the performance of the policy $\hat{\pi}$ in the following episodes. We study the total regret of $\mathcal{ALG}_2$ from episode $K+1$ to episode $K_1$.

Recently, Zhou et al. (2020a) proposed the following regret lower bound for linear mixture MDPs. In order to avoid confusion, we use the notation $K_2$ instead of $K$ to denote the number of total episodes of the regret minimization problem.

**Lemma 16.** *(Theorem 5.6 in Zhou et al. (2020a)) Let $B > 1$ and suppose $K_2 \geq \max\left\{(d-1)^2H/2, (d-1)/(32H(B-1))\right\}$, $d \geq 4$, $H \geq 3$. Then for any algorithm there exists an episodic, B-bounded linear mixture MDP parameterized by $\Theta = (\theta_1, \cdots, \theta_H)$ such that the expected regret is at least $\Omega(dH\sqrt{HK_2})$.*

To prove the above regret lower bound, Zhou et al. (2020a) construct a class of hard instances which is an extension of the hard instance class for linear bandits (cf. Theorem 24.1 in Lattimore & Szepesvári (2020)). They show that for any algorithm, there exists a hard instance in the instance class such that the regret is at least $\Omega(dH\sqrt{HK_2})$. By Theorem 16 and setting $K_2 = K_1$, we know that for any possible algorithm $\mathcal{ALG}_2$, there exists a hard instance $\mathcal{M}$ such that $\sum_{k=1}^{K_1} \mathbb{E}\left(V^*(s_1) - V^{\pi_k}(s_1)\right) \geq c_2 dH\sqrt{HK_1}$ for a positive constant $c_2$, where $\pi_k$ denotes the policy used in episode $k$ for algorithm $\mathcal{ALG}_2$. The expectation is over all randomness of the algorithm and the environment.

Note that in the hard instance constructed in Zhou et al. (2020a), the per-step regret is at most $\mathbb{E}\left(V^*(s_1) - V^{\pi_k}(s_1)\right) \leq \frac{dH}{4\sqrt{2}}\sqrt{\frac{H}{K_1}}$ for any episode $k \leq K_1$. By choosing $c = \max\left\{\frac{1}{2\sqrt{2}c_2}, 2\right\}$, we know that for the instance $\mathcal{M}$,

$$\sum_{k=K_1/c+1}^{K_1} \mathbb{E}\left(V^*(s_1) - V^{\pi_k}(s_1)\right) \geq \left(c_2 - \frac{1}{4\sqrt{2}c}\right)dH\sqrt{HK_1} \geq \frac{c_2}{2}dH\sqrt{HK_1}. \tag{204}$$

By the definition of $\mathcal{ALG}_2$, we have $\pi_k = \hat{\pi}$ if $k > K_1/c$. Therefore, we have

$$(c-1)K\mathbb{E}_{\hat{\pi}\sim\mu_\pi, s_1\sim\mu}\left(V^*(s_1) - V^{\hat{\pi}}(s_1)\right) \geq \frac{c_2}{2}dH\sqrt{HK_1} = \frac{c_2 c}{2}dH\sqrt{HK}. \tag{205}$$

Dividing both sides by $(c-1)K$, we have

$$\mathbb{E}_{\hat{\pi}\sim\mu_\pi, s_1\sim\mu}\left(V^*(s_1) - V^{\hat{\pi}}(s_1)\right) \geq \frac{c_2 c}{2(c-1)}dH\sqrt{H/K}. \tag{206}$$

The above inequality indicates that, for any algorithm $\mathcal{ALG}_1$, there exists an instance $\mathcal{M}$ such that the expected sub-optimality gap of the policy returned by $\mathcal{ALG}_1$ is at least $\frac{c_2 c}{2(c-1)}dH\sqrt{H/K}$ after collecting samples for $K$ episodes.

Suppose $\mathcal{ALG}_1$ returns an $\epsilon$-optimal policy $\hat{\pi}$ with probability at least $1 - \delta$. Recall that the per-step sub-optimality gap is at most $\mathbb{E}\left(V^*(s_1) - V^{\pi_k}(s_1)\right) \leq \frac{dH}{4\sqrt{2}}\sqrt{\frac{H}{cK}}$ in the constructed hard instance. We have

$$(1-\delta)\cdot\epsilon + \delta\cdot\frac{dH}{4\sqrt{2}}\sqrt{\frac{H}{cK}} \geq \mathbb{E}_{\hat{\pi}\sim\mu_\pi, s_1\sim\mu}\left(V^*(s_1) - V^{\hat{\pi}}(s_1)\right) \geq \frac{c_2 c}{2(c-1)}dH\sqrt{H/K}. \tag{207}$$

We set $\delta$ to be a constant satisfying $0 < \delta < \min\left\{1, \frac{2\sqrt{2}c^{1.5}c_2}{c-1}\right\}$. Solving the above inequality, we have $K \geq \frac{Cd^2H^3}{\epsilon^2}$ for a positive constant $C$. □

# D    IMPROVED BOUND FOR REWARD-FREE EXPLORATION IN LINEAR MDPS

In this section, we explain how our choice of the exploration-driven reward can be used to improve the sample complexity bound for reward-free exploration in linear MDPs (Wang et al., 2020). We study the same reward-free setting as that in Wang et al. (2020), which is briefly explained in Appendix D.1. We describe our algorithm and bound in Appendix D.2, and prove our theorem in Appendix D.3.

## D.1    PRELIMINARIES

For the completeness of explanation, we briefly restate the reward-free setting studied in Wang et al. (2020). Compared with the setting in this work, the main differences are twofold: Firstly, they study the linear MDPs setting instead of linear mixture MDPs in this work. Secondly, they study the standard reward-free exploration setting without the constraints of the plug-in solver.

The linear MDP assumption, which was first introduced in Yang & Wang (2019); Jin et al. (2020b), states that the model of the MDP can be represented by linear functions of given features.

**Assumption 1.** *(Linear MDP) an MDP $\mathcal{M} = (\mathcal{S}, \mathcal{A}, P, R, H, \nu)$ is said to be a linear MDP if the following hold:*

- *There are $d$ unknown signed measures $\mu_h = (\mu_h^{(1)}, \mu_h^{(2)}, \cdots, \mu_h^{(d)})$ such that for any $(s, a, s') \in \mathcal{S} \times \mathcal{A} \times \mathcal{S}$, $P_h(s' \mid s, a) = \langle \mu_h(s'), \phi(s, a)\rangle$.*

- *There exists $H$ unknown vectors $\eta_1, \eta_2, \cdots, \eta_H \in \mathbb{R}^d$ such that for any $(s, a) \in \mathcal{S} \times \mathcal{A}$, $R_h(s, a) = \langle \phi(s, a), \eta_h\rangle$.*

*We assume for all $(s, a) \in \mathcal{S} \times \mathcal{A}$ and $h \in [H]$, $\|\phi(s, a)\| \leq 1$, $\|\mu_h(s)\|_2 \leq \sqrt{d}$ and $\|\eta\|_2 \leq \sqrt{d}$.*

For the reward-free exploration studied in Wang et al. (2020), the agent can collects a dataset of visited state-action pairs $\mathcal{D} = \left\{ \left( s_h^k, a_h^k \right) \right\}_{(k,h) \in [K] \times [H]}$ which will be used in the planning phase. Then during the planning phase, the agent can follow a certain designed learning algorithm to calculate an $\epsilon$-optimal policy w.r.t. the reward function $R$ using the dataset $\mathcal{D}$.

### D.2 ALGORITHM AND THEOREM

Our algorithm can be divided into two parts. The exploration phase of the algorithm is presented in Algorithm 3, and the planning phase is presented in Algorithm 4.

---

**Algorithm 3** Reward-free Exploration for Linear MDPs: Exploration Phase

---

Input: Failure probability $\delta > 0$ and target accuracy $\epsilon > 0$
$\beta \leftarrow c_\beta \cdot dH \sqrt{\log \left( dH \delta^{-1} \varepsilon^{-1} \right)}$ for some $c_\beta > 0$
$K \leftarrow c_K \cdot d^3 H^4 \log \left( dH \delta^{-1} \varepsilon^{-1} \right) / \varepsilon^2$ for some $c_K > 0$
**for** episode $k = 1, 2, \cdots, K$ **do**
5:     $Q_{k,H+1}(\cdot, \cdot) \leftarrow 0$, $V_{k,H+1}(\cdot) \leftarrow 0$
      **for** step $h = H, H-1, \cdots, 1$ **do**
         $\Lambda_{k,h} \leftarrow \sum_{t=1}^{k-1} \phi(s_{t,h}, a_{t,h}) \phi(s_{t,h}, a_{t,h})^\top + I$
         $u_{k,h}(\cdot, \cdot) \leftarrow \beta \sqrt{\phi(\cdot, \cdot)^\top \left( \Lambda_{k,h} \right)^{-1} \phi(\cdot, \cdot)}$
         Define the exploration-driven reward function $R_{k,h}(\cdot, \cdot) = u_{k,h}(\cdot, \cdot)$
10:        $\hat{w}_{k,h} \leftarrow (\Lambda_{k,h})^{-1} \sum_{t=1}^{k-1} \phi(s_{t,h}, a_{t,h}) V_{t,h+1}(s_{t,h+1})$
         $Q_{k,h}(\cdot, \cdot) \leftarrow \min \left\{ \hat{w}_{k,h}^\top \phi(\cdot, \cdot) + R_{k,h}(\cdot, \cdot) + u_{k,h}(\cdot, \cdot), H \right\}$
         $V_{k,h}(s) \leftarrow \max_{a \in \mathcal{A}} Q_{k,h}(s, a)$, $\pi_{k,h}(s) = \arg\max_{a \in \mathcal{A}} Q_{k,h}(s, a)$
      **end for**
      **for** step $h = 1, 2, \cdots, H$ **do**
15:        Take action $a_{k,h} = \pi_{k,h}(s_{k,h})$ and observe $s_{k,h+1} \sim P_h(s_{k,h}, a_{k,h})$
      **end for**
**end for**
Output: $\mathcal{D} \leftarrow \left\{ (s_{k,h}, a_{k,h}) \right\}_{(k,h) \in [K] \times [H]}$.

---

**Algorithm 4** Reward-free Exploration for Linear MDPs: Planning Phase

---

Input: Dataset $\mathcal{D} \leftarrow \left\{ (s_{k,h}, a_{k,h}) \right\}_{(k,h) \in [K] \times [H]}$, reward functions $R = \{R_h\}_{h \in [H]}$
$Q_{k,H+1}(\cdot, \cdot) = 0$, $V_{k,H+1}(\cdot) = 0$
**for** step $h = H, H-1, \cdots, 1$ **do**
   $\Lambda_h \leftarrow \sum_{t=1}^K \phi(s_{t,h}, a_{t,h}) \phi(s_{t,h}, a_{t,h})^\top + I$
5:   $u_h(\cdot, \cdot) \leftarrow \min \left\{ \beta \sqrt{\phi(\cdot, \cdot)^\top \left( \Lambda_{k,h} \right)^{-1} \phi(\cdot, \cdot)}, H \right\}$
   $\hat{w}_h \leftarrow (\Lambda_h)^{-1} \sum_{t=1}^K \phi(s_{t,h}, a_{t,h}) V_{t,h+1}(s_{t,h+1})$
   $Q_h(\cdot, \cdot) \leftarrow \min \left\{ \hat{w}_h^\top \phi(\cdot, \cdot) + R_h(\cdot, \cdot) + u_h(\cdot, \cdot), H \right\}$
   $V_h(s) \leftarrow \max_{a \in \mathcal{A}} Q_h(s, a)$, $\pi_h(s) = \arg\max_{a \in \mathcal{A}} Q_h(s, a)$
**end for**
10: Output: $\pi = \{\pi_h\}_{h \in [H]}$

---

Compared with the algorithm in Wang et al. (2020), the main difference is that we set $R_{k,h} = u_{k,h}$ in the exploration phase, instead of $R_{k,h} = u_{k,h}/H$. The following theorem states the complexity bound of our algorithms.

**Theorem 5.** *With probability at least $1 - \delta$, after collecting $O(d^3 H^4 \log(dH\delta^{-1}\epsilon^{-1})/\epsilon^2)$ trajectories during the exploration phase, our algorithm outputs an $\epsilon$-optimal policy for any given reward during the planning phase.*

### D.3 PROOF OF THEOREM 5

The proof of Theorem 5 follows the proof framework in Wang et al. (2020) with a slight modification. Therefore, we only sketch the proof and mainly focus on explaining the differences. Firstly, we introduce the value function $\tilde{V}_h^*(s, R)$, which is recursively defined from step $H + 1$ to step 1:

$$\tilde{V}_{H+1}^*(s, R) = 0, \forall s \in \mathcal{S} \tag{208}$$

$$\tilde{V}_h^*(s, R) = \max_{a \in \mathcal{A}} \left\{ \min \left\{ R_h(s, a) + P_h \tilde{V}_{h+1}^*(s, a, R), H \right\} \right\}, \forall s \in \mathcal{S}, h \in [H] \tag{209}$$

Compared with the definition of $V_h^*(s, R)$, the main difference is that we take minimization over the value and $H$ at each step. We can similarly define $\tilde{Q}_h^*(s, a, R)$, $\tilde{V}_h^\pi(s, R)$ and $\tilde{Q}_h^\pi(s, a, R)$.

**Lemma 17.** *With probability $1 - \delta/2$, for all $k \in [K]$,*

$$\tilde{V}_1^*(s_{1,k}, R_k) \leq V_{k,1}(s_{1,k}) \tag{210}$$

*and*

$$\sum_{k=1}^K V_{k,1}(s_{k,1}) \leq c\sqrt{d^3 H^4 K \log(dKH/\delta)} \tag{211}$$

*for some constant $c > 0$ where $V_{k,1}$ is as defined in Algorithm 3.*

This lemma corresponds to Lemma 3.1 in Wang et al. (2020). The main difference is that we replace $V_1^*$ with $\tilde{V}_1^*$. Note that $\tilde{V}_h^*(s, R) \leq H$ by the definition of $\tilde{V}_h^*(s, R)$. Lemma 17 can be similarly proved following the proof of Lemma 3.1 in Wang et al. (2020) and replacing $V_h^*$ by $\tilde{V}_h^*$ during the proof.

**Lemma 18.** *With probability $1 - \delta/4$, for the function $u_h(\cdot, \cdot)$ defined in Line 5 in Algorithm 4, we have*

$$\mathbb{E}_{s \sim \mu}\left[\tilde{V}_1^*(s, u_h)\right] \leq c'\sqrt{d^3 H^4 \cdot \log(dKH/\delta)/K} \tag{212}$$

Compared with Lemma 3.2 in Wang et al. (2020), we replace the term $V_1^*(s, u_h/H)$ with $\tilde{V}_1^*(s, u_h)$. Note that with our choice of exploration-driven reward in the exploration phase (i.e. $R_{k,h} = u_{k,h}$), we replace the term $u_h/H$ with $u_h$ in the expectation. This lemma can be proved by following the proof of Lemma 3.2 in Wang et al. (2020).

**Lemma 19.** *With probability $1 - \delta/2$, for any reward function satisfying Assumption 1 and all $h \in [H]$, we have*

$$Q_h^*(\cdot, \cdot, r) \leq Q_h(\cdot, \cdot) \leq R_h(\cdot, \cdot) + \sum_{s'} P_h(s' \mid \cdot, \cdot) V_{h+1}(s') + 2u_h(\cdot, \cdot). \tag{213}$$

Since our algorithm in the planning phase is exactly the same with that of Wang et al. (2020), this lemma shares the same idea of Lemma 3.3 in Wang et al. (2020).

Now we can prove Theorem 5 with the help of Lemma 18 and Lemma 19.

*Proof.* (Proof of Theorem 5) We condition on the events defined in Lemma 18 and Lemma 19, which hold with probability at least $1 - \delta$. By Lemma 19, we have for any $s \in \mathcal{S}$,

$$V_1(s) = \max_s Q_1(s, a) \geq \max_a Q_1^*(s, a, R) = V_1^*(s, R), \tag{214}$$

which implies

$$\mathbb{E}_{s_1 \sim \mu}\left[V_1^*(s_1, R) - V_1^\pi(s_1, r)\right] \leq \mathbb{E}_{s_1 \sim \mu}\left[V_1(s_1) - V_1^\pi(s_1, R)\right]. \tag{215}$$

Note that $0 \leq V_h(s) \leq H$ and $0 \leq V_h^\pi(s, R) \leq H$ since $0 \leq R(s, a) \leq 1$. Therefore, we always have $V_h(s) - V_h^\pi(s, R) \leq H$. For any $s_h$, we have

$$V_h(s_h) - V_h^\pi(s_h, R) \tag{216}$$

$$\leq \min\left\{H, P_h V_{h+1}(s_h, \pi_h(s_h)) + 2u_h(s_h, \pi_h(s_h)) - P_h V_{h+1}^\pi(s_{h+1}, R)\right\} \tag{217}$$

By recursively decomposing $V_h(s_h) - V_h^\pi(s_h, R)$ from step $H$ to 1, we have

$$\mathbb{E}_{s_1 \sim \mu}[V_1(s_1) - V_1^\pi(s_1, r)] \leq \mathbb{E}_{s \sim \mu}\left[\tilde{V}_1^\pi(s, u)\right]. \tag{218}$$

By definition of $\tilde{V}_1^*(s, u)$, we have $\mathbb{E}_{s \sim \mu}\left[\tilde{V}_1^\pi(s, u)\right] \leq 2\mathbb{E}_{s \sim \mu}\left[\tilde{V}_1^*(s, u)\right]$. By Lemma 18,

$$\mathbb{E}_{s \sim \mu}\left[\tilde{V}_1^*(s, u)\right] \leq c'\sqrt{d^3 H^4 \cdot \log(dKH/\delta)/K}. \tag{219}$$

By taking $K = c_K d^3 H^4 \log(dH\delta^{-1}\epsilon^{-1})/\epsilon^2$ for a sufficiently large constant $c_K > 0$, we have

$$\mathbb{E}_{s_1 \sim \mu}[V_1^*(s_1, R) - V_1^\pi(s_1, r)] \leq c'\sqrt{d^3 H^4 \cdot \log(dKH/\delta)/K} \leq \epsilon. \tag{220}$$

$\square$

# E    AUXILIARY LEMMAS

**Lemma 20.** *(Self-Normalized Bound for Vector-Valued Martingales, Theorem 1 and 2 in Abbasi-Yadkori et al. (2011)) Let $\{F_t\}_{t=0}^\infty$ be a filtration. Let $\{\eta_t\}_{t=1}^\infty$ be a real-valued stochastic process such that $\eta_t$ is $F_t$-measurable and $\eta_t$ is conditionally $R$-sub-Gaussian for some $R \geq 0$, i.e.*

$$\forall \lambda \in \mathbb{R}, \mathbb{E}[e^{\lambda \eta_t}|F_{t-1}] \leq \exp\left(\frac{\lambda^2 R^2}{2}\right). \tag{221}$$

*Let $\{X_t\}_{t=1}^\infty$ be an $\mathbb{R}^d$-valued stochastic process such that $X_t$ is $F_{t-1}$-measurable. Assume that $V$ is a $d \times d$ positive definite matrix. For any $t \geq 0$, define*

$$\bar{V}_t = V + \sum_{s=1}^t X_s X_s^\top, S_t = \sum_{s=1}^t \eta_s X_s. \tag{222}$$

*Then, for any $\delta > 0$, with probability at least $1 - \delta$, for all $t \geq 0$,*

$$\|S_t\|_{\bar{V}_t^{-1}}^2 \leq 2R^2 \log\left(\frac{\det(\bar{V}_t)^{1/2}\det(V)^{-1/2}}{\delta}\right). \tag{223}$$

*Further, let $V = I\lambda, \lambda > 0$. Define $Y_t = \langle X_t, \theta_* \rangle + \eta_t$ and assume that $\|\theta_*\|_2 \leq S, \|X_t\|_2 \leq L, \forall t \geq 1$. Then for any $\delta > 0$, with probability at least $1 - \delta$, for all $t \geq 0$, $\theta_*$ satisfies*

$$\left\|\hat{\theta}_t - \theta_*\right\|_{\bar{V}_t} \leq R\sqrt{d\log\left(\frac{1 + tL^2/\lambda}{\delta}\right)} + \lambda^{1/2}S, \tag{224}$$

*where $\hat{\theta}_t$ is the $l^2$-regularized least-squares estimation of $\theta_*$ with regularization parameter $\lambda > 0$ based on history samples till step $t$.*

**Lemma 21.** *(Lemma 11 in Abbasi-Yadkori et al. (2011)) Let $\{X_t\}_{t=1}^\infty$ be a sequence in $\mathbb{R}^d$, $V$ a $d \times d$ positive definite matrix and define $\bar{V}_t = V + \sum_{s=1}^t X_s X_s^\top$. Then, we have that*

$$\log\left(\frac{\det(\bar{V}_n)}{\det(V)}\right) \leq \sum_{t=1}^n \|X_t\|_{\bar{V}_{t-1}^{-1}}^2. \tag{225}$$

*Further, if $\|X_t\|_2 \leq L$ for all $t$, then*

$$\sum_{t=1}^n \min\left\{1, \|X_t\|_{\bar{V}_{t-1}^{-1}}^2\right\} \leq 2\left(\log\det(\bar{V}_n) - \log\det V\right) \leq 2\left(d\log\left(\left(\text{trace}(V) + nL^2\right)/d\right) - \log\det V\right), \tag{226}$$

*and finally, if $\lambda_{\min}(V) \geq \max(1, L^2)$, then*

$$\sum_{t=1}^n \|X_t\|_{\bar{V}_{t-1}^{-1}}^2 \leq 2\log\frac{\det(\bar{V}_n)}{\det(V)}. \tag{227}$$

**Lemma 22.** *(Bernstein inequality for vector-valued martingales, Theorem 4.1 in Zhou et al. (2020a)) Let $\{\mathcal{G}_t\}_{t=1}^{\infty}$ be a filtration, $\{x_t, \eta_t\}_{t \geq 1}$ a stochastic process so that $x_t \in \mathbb{R}^d$ is $\mathcal{G}_t$-measurable and $\eta_t \in \mathbb{R}$ is $\mathcal{G}_{t+1}$-measurable. Fix $R, L, \sigma, \lambda > 0, \mu^* \in \mathbb{R}^d$. For $t \geq 1$, let $y_t = \langle \mu^*, x_t \rangle + \eta_t$ and suppose that $\eta_t, x_t$ also satisfy*

$$|\eta_t| \leq R, \mathbb{E}\left[\eta_t \mid \mathcal{G}_t\right] = 0, \mathbb{E}\left[\eta_t^2 \mid \mathcal{G}_t\right] \leq \sigma^2, \|x_t\|_2 \leq L. \tag{228}$$

*Then, for any $0 < \delta < 1$, with probability at least $1 - \delta$, we have*

$$\forall t > 0, \left\| \sum_{i=1}^t x_i \eta_i \right\|_{z_t^{-1}} \leq \beta_t, \|\mu_t - \mu^*\|_{z_t} \leq \beta_t + \sqrt{\lambda} \|\mu^*\|_2, \tag{229}$$

*where for $t \geq 1$, $\mu_t = Z_t^{-1} b_t$, $Z_t = \lambda I + \sum_{i=1}^t x_i x_i^\top$, $b_t = \sum_{i=1}^t y_i x_i$ and*

$$\beta_t = 8\sigma \sqrt{d \log\left(1 + tL^2/(d\lambda)\right) \log\left(4t^2/\delta\right)} + 4R \log\left(4t^2/\delta\right). \tag{230}$$

