# OpenReview forum: "Near-Optimal Reward-Free Exploration for Linear Mixture MDPs with Plug-in Solver"
_ICLR.cc/2022/Conference — ICLR 2022 Spotlight_

### Official Review · Reviewer_kLwC · 2021-10-22

**Correctness:** 4
**Technical Novelty And Significance:** 3
**Empirical Novelty And Significance:** 1
**Recommendation:** 8
**Confidence:** 3

**Main Review:**

After Response Update

In their replies, the authors have clarified my concerns regarding the lower bound and the computational tractability. They are incorporating useful changes in the paper, including a comparison with a relevant concurrent work. I am now fully convinced this is a valuable paper, and I am increasing my score accordingly.


-------

I provide below some detailed comments and doubts that the authors might address in their response.

COMPUTATIONAL TRACTABILITY

I think the paper is somewhat missing an assessment of the tractability of the propose methods, and especially the computational complexity of the optimization problems involved. Can the authors clarify this point?

LOWER BOUND

The paper provides a lower bound $\Omega (H^3 d^2 \epsilon^{-2})$ by reducing a previous result from tabular MDPs, i.e., $\Omega (H^2 S^2 A \epsilon^{-2})$ (Jin et al., 2020a). Can the authors clarify how this reduction is carried out? I would say that tabular MDPs can be obtained from a linear mixture MDPs with $d = S^2 A$. I see why there is an additional $H$ factor w.r.t. time-homogenous settings, but where is the additional $d$ factor coming from?

COMPARING SUB-OPTIMALITY

The sub-optimality guarantee of Theorems 1, 2 includes an additional $\epsilon_{opt}$ term with respect to previous works, which instead provide $\epsilon$-optimality guarantees. I am wondering if this should be intended as an unavoidable cost of the generality of the proposed method, which works with any planning solver, or there is a way to make a fair comparison with the complexity rates of previous works.

RELATED WORK

The work (W. Zhang et al., Reward-free model-based reinforcement learning with linear function approximation, 2021) seems to be very much related to the setting of this paper. Even if this is very recent, and thus concurrent to this work, can the authors provide a brief discussion over similarities and differences of their approaches?

ADDITIONAL REFERENCES

Reward-free RL is a relatively novel field, which does not count a huge number of works yet, thus I would suggest the authors to try to reference all of them, even if they are not crucially relevant or comparable to the proposed setting. E.g., they could consider adding:
- Z. Zhang et al., Near optimal reward-free reinforcement learning, 2021;
- W. Zhang et al., Reward-free model-based reinforcement learning with linear function approximation, 2021;
- Qiu et al., On reward-free RL with kernel and neural function approximations, 2021;

as well as works in the close task-agnostic setting:
- X. Zhang and Singla, Task-agnostic exploration in reinforcement learning, 2020;
- Wu et al., Gap-dependent unsupervised exploration for reinforcement learning, 2021;

and multi-agent settings:
- Bai and Jin, Provable self-play algorithms for competitive reinforcement learning, 2021;
- Liu et al., A sharp analysis of model-based reinforcement learning with self-play, 2021.

**Summary Of The Paper:**

This paper addresses the reward-free exploration problem with function approximation under linear mixture MDP assumption. It proposes a pair of model-based exploration algorithms, one with a lightweight methodology and sample complexity rate $\tilde{O}(H^4 d^2 \epsilon^{-2})$ and the other as a refined and more convoluted version with rate $\tilde{O}(H^3 d^2 \epsilon^{-2})$, which is matching the reported lower bound up to logarithmic factors. Crucially, the proposed approach can work with any planning solver to provide an $(\epsilon + \epsilon_{opt})$-optimal policy for any reward function.

**Summary Of The Review:**

This works looks like a solid contribution to the reward-free RL literature. To the best of my knowledge, it provides the best known sample complexity rate for reward-free exploration under linear mixture assumptions (especially, it improves over W. Zhang et al. 2021 by a factor of H), and the best rate for a reward-free method that can work with any planning solver (e.g., Jin et al. 2020a, W. Zhang et al. 2021).

Since I believe these improvements are valuable, I am providing a positive evaluation for this work. A careful inspection of the novelty of the analysis (especially w.r.t. Zhou et al. 2020a), as well as additional reassurances from the authors on some questions over the computational tractability of the proposed algorithms and the sample complexity lower bound, could make me consider raising my score towards an even stronger accept.

I believe that extending the proposed analysis to improve known sample complexity rates under the looser linear MDP assumption, as the authors suggest in Section 4.2, would potentially increase the value of this work from solid to great.

---

> ### Author Response · Authors · 2021-11-20
> **Response to Reviewer 3**
>
> We thank Reviewer 3 for the positive assessment and insightful comments.
>
> Regarding the computational issue: Thanks for the suggestions. We have added the implementation details of Algorithm 1 in Appendix A.2. Overall, the optimization problem defined in line 8 can be relaxed as an linear programming problem. However, the sample complexity will be worse by a factor of $d$ if we solve the relaxed problem as an approximation. The optimization problem defined in line 19 can be reformulated as a quadratic programming problem, which can also be solved by standard optimization approaches. As for Algorithm 2, the optimization problem defined in line 7 cannot be solved efficiently yet. It is an interesting future problem to design algorithms that are both statistically optimal and computationally efficient.
>
> Regarding the lower bound: Thanks for pointing out this issue. We understand that the direct reduction cannot lead to the tight lower bound in linear setting. Therefore, we propose a new lower bound of $\Omega(d^2H^3/\epsilon^2)$ (Theorem 3) for the reward-free exploration setting. We achieve this by connecting the sample complexity lower bound with the regret lower bound of certain constructed learning algorithms in the standard online exploration setting. Please see Appendix C for the detailed discussion.
>
> Regarding the sub-optimality: The introduction of planning error $\epsilon_{opt}$ is to make our results more general for any possible planning algorithms in the planning phase. This error can be $0$ if we use value iteration algorithms similarly with the previous reward-free works (e.g. Zhang et al. 2021) in the planning phase. If the state space is extremely large and the computation cost of the value iteration algorithms are unacceptable, we can use $n$-step bootstrapping planners or decision-time planning algorithms to find a good policy. However, the planning error $\epsilon_{opt}$ is unavoidable in this case.
>
> Regarding the related work: Thanks for introducing the related literature of reward-free exploration. We have updated our related work section accordingly. There are mainly three differences between our results and the recent work of Zhang et al. Firstly, we propose the setting of the reward-free exploration with plug-in solver, which is a more general setting that covers the standard reward-free exploration studied in the previous results including Zhang et al. This setting is also more difficult than that studied in Zhang et al. since the agent can only observe the transition estimation in the planning phase and our results holds for any possible planning algorithms. Secondly, our sample complexity bounds are tighter than theirs by a factor of $H^2$ (Note that they study time-homogeneous MDPs setting while we focus on time-inhomogeneous case. Our bound can be further improved by a factor of $H$ when transformed to their setting). Roughly speaking, this is achieved by setting the exploration-driven reward $R_{k,h}(s,a) = u_{k,h}(s,a)$ instead of $u_{k,h}(s,a)/H$ and a more careful analysis to bound the model estimation error in the planning phase. Thirdly, our lower bound is tighter than theirs in the dependence on $d$, which indicates that our sample complexity bound is statistically optimal except for logarithmic factors.
>
> Regarding the improved bound in linear MDP setting: Thanks for your suggestions. We have added the detailed explanation of how our techniques can improve the sample complexity in linear MDPs in Appendix D. The improved bound is stated in Theorem 5.

---

> > ### Comment · Reviewer_kLwC · 2021-11-26
> > **After Response**
> >
> > I would like to thank the authors for their thorough replies and their remarkable effort in improving the paper according to reviewers' suggestions.
> >
> > The only additional doubts that I had, related to the sample complexity lower bound, are well addressed in the discussion with Reviewer ZcuD.
> >
> > As a final note, I would suggest the authors to mention the main aspects of Appendix A.2 in the main paper. Assuming the authors will incorporate this change, I am updating my score towards a clearer accept.

---

> > > ### Author Response · Authors · 2021-11-27
> > > **Thanks**
> > >
> > > Thanks for your support. We appreciate the suggestions from all the reviewers during the review period. We will address all the comments, including fixing typos, clarifying definitions and statements, and moving implementation details to the main paper in the next version.

---

### Official Review · Reviewer_ZcuD · 2021-11-02

**Correctness:** 3
**Technical Novelty And Significance:** 3
**Empirical Novelty And Significance:** 3
**Recommendation:** 6
**Confidence:** 3

**Main Review:**

Overall, this paper makes a fundamental contribution to the reward-free exploration field. The proposed solution to the technical challenge of the Bernstein-type analysis under this scenario is novel. Moreover, the paper is well organized.

However, several issues/limitations preclude me from fully accepting this paper.

1. The claim of “near-optimal” does not appear suitable in my opinion, provided that (Jin et al., 2020) considered the tabular MDP. Can you explain more why this is a suitable argument? In particular, why can we translate the $S^2A$ term to $d^2$? To my understanding, the reduction between tabular and linear settings is strange/unsafe.  To see this, if we let $d=S^2 A$ (i.e.,  use the one-hot feature as the linear feature), the resultant upper bound $d^2$ becomes $O(S^{4}A^2)$. This upper bound (actually this translation) is not tight, compared with the lower bound in the tabular setting.

I would suggest, if the authors agree, to adjust the language to be clear of this mismatch in problem settings between this paper and (Jin et al., 2020). This would make this paper’s place in the literature more clear.

2. Under the linear mixture MDPs setting, there is no concrete optimization procedure for the $\epsilon_\text{opt}$-optimal model-based solver (a key assumption in this paper). This assumption can be well addressed if the state space is finite (see (Jin et al., 2020)). I do not aware of efficient optimization procedures when the state space is infinite. Can you provide some algorithms that satisfy this assumption when the state space is infinite?  I have the same question for several optimization steps in the algorithm design; for example, Line 9 in Algorithm 1 and Line 7 in Algorithm 2.

This issue would not significantly detract from the contribution of this paper, but it would be better if you could provide some possible solutions or provide some discussion.

3. The sample complexity in Theorem 1 and Theorem 2 does not show the dependence on $B$ (i.e., the upper bound for the weight norm of $\theta$ in Definition 1). Indeed, we cannot treat $B$ as a absolute constant because $B$ contains the state and action space size information. For instance, considering the one-hot feature $\phi$, we have that $d=SAS$ and $\Vert \phi_{V}(s, a) \Vert_2 \leq \sqrt{S}$ based on the definition $\phi_{V}(s, a) = \sum_{s^\prime \in \mathcal{S}} \phi(s, a, s^\prime) V(s^\prime)$. This contradicts with the assumption that $\Vert \phi_{V}(s, a) \Vert_2 \leq 1$. To resolve this issue, we can normalize the one-hot feature by dividing $1/\sqrt{S}$. Consequently, to ensure that $P_h(s^\prime | s, a) = \theta_h^{\top} \phi(s, a, s^\prime)$ is a valid transition, we could set $B = \sqrt{S^2 A}$.

I realize that previous literature somehow does not show the dependence on $B$. But it’s better to discuss this point, which could make the connection between tabular and linear settings more clear. Please correct me if my understanding is wrong.

4. Some important steps in proposed algorithms are not well discussed. For instance, how to find a valid transition at Line 20 in Algorithm 1. A similar issue holds for Line 5 in Algorithm 2. I would suggest the authors involve more implementation details.


Several sentences are confusing/misleading in this submission. I suggest revising these statements to improve clarity.

1. “Compared to our results, the standard plug-in approach (Agarwal et al., 2020; Cui & Yang, 2020) requires reward signals to guide exploration”

There is no exploration issue in Agarwal et al., 2020.

2. “Based on the Berstein inequality for vector-valued martingales”

There is a typo: Berstein -> Bernstein

3. “Compared with the reward-free setting studied in the previous literature ...., the main difference is that we require that the algorithm maintains an model estimation instead of all the history samples after the exploration phase”

There is a typo: an model estimation -> a model estimation.

4. “Then with probability at least 1 − $\delta$, for any given reward”

In Theorem 1 and Theorem 2, “Then with probability 1-$\delta$” should be posted before the first claim or the authors should clarify the randomness source.

5. The paragraph below Lemma 3 seems incomplete since no conclusion is given. As a result, readers must combine Lemma 3 and Lemma 1 to infer the result that the authors want to claim.

6. $R_K$ is not properly defined in (4).  Maybe you mean the concatenation of all one-step “exploration rewards” in iteration $K$?


I would like to raise my score if the above concerns are addressed in the authors’ response.

---

During the review, I noticed that there is a related paper in the Arxiv.

Zhang, Weitong, Dongruo Zhou, and Quanquan Gu. "Reward-Free Model-Based Reinforcement Learning with Linear Function Approximation." arXiv preprint arXiv:2110.06394 (2021).  https://arxiv.org/abs/2110.06394

According to the review guide, authors are excused for such a paper (i.e., the Arxiv paper would not affect my decision). But, I encourage authors to incorporate the following points to further improve the quality of this paper:
1. Compare Algorithms 1 and 2 in this work with algorithms in (Zhang et al., 2021).
2. Point out which steps/components/analyses in this paper lead to better results.
3. Discuss involved optimization steps as Zhang et al. do.


===========================

After rebuttal, the authors have addressed my concern regarding the lower bound and computational details. Thus, I raise my score from 5 to 6.







**Summary Of The Paper:**

This paper studies reward-free exploration with linear function approximation (i.e., under the setting of linear mixture MDPs). Specifically, this work introduces two algorithms: the first one has a simple form with the sample complexity $\tilde{\mathcal{O}}(d^2H^4 / \varepsilon^2)$, and the second one is much complicated (i.e., with 5 estimators) but has an improved sample complexity $\tilde{\mathcal{O}}(d^2H^3 / \varepsilon^2)$. To improve the dependence on $H$, the main technical challenge is that the traditional Bernstein-type analysis could not be directly applied because the variance summation of $\{\tilde{V}\}_{h \in [H]}$ is not induced from the same policy and transition function.  This paper addresses this technical challenge and makes a solid contribution to reward-free exploration.

**Summary Of The Review:**

This paper improves the statistical sample complexity (in terms of $H$) for reward-free exploration with linear function approximation. Thus, this paper deserves to be accepted and I would raise my score if the mentioned concerns are addressed.

---

> ### Author Response · Authors · 2021-11-20
> **Response to Reviewer 2**
>
> We thank Reviewer 2 for the detailed review and insightful comments.
>
> Regarding the lower bound: Thanks for pointing out this issue. We understand that the direct reduction cannot lead to the tight lower bound in linear setting. Therefore, we propose a new lower bound of $\Omega(d^2H^3/\epsilon^2)$ (Theorem 3) for the reward-free exploration setting. We achieve this by connecting the sample complexity lower bound with the regret lower bound of certain constructed learning algorithms in the standard online exploration setting. Please see Appendix C for the detailed discussion.
>
> Regarding the model-based solver: The introduction of the model-based solver is to make our results more general for any possible planning algorithms in the planning phase. For example, our results still hold if we use value iteration algorithms similarly with the previous reward-free works (e.g. Zhang et al. 2021)  as the planning solver, in which case the planning error $\epsilon_{opt} = 0$. If the state space is extremely large or even infinite, we can use $n$-step bootstrapping planners or decision-time planning algorithms to find a good policy. However, the planning error $\epsilon_{opt}$ is unavoidable in this case.
>
> Regarding the dependence on $B$: Thanks for the suggestions. We have included the explicit dependence on $B$ in the algorithm, the theorem and the proof. Our sample complexity has only logarithmic dependence on $B$.
>
> Regarding the implementation details: Thanks for the suggestions. We have added the implementation details of Algorithm 1 in Appendix A.2. Overall, the optimization problem defined in line 8 can be relaxed as an linear programming problem. However, the sample complexity will be worse by a factor of $d$ if we solve the relaxed problem as an approximation. The optimization problem defined in line 19 can be reformulated as a quadratic programming problem, which can also be solved by standard optimization approaches. As for Algorithm 2, the optimization problem defined in line 7 cannot be solved efficiently yet. It is an interesting future problem to design algorithms that are both statistically optimal and computationally efficient.
>
> Regarding the Comparison with Zhang et al.: Thanks for introducing the concurrent work of Zhang et al. There are mainly three differences between our results and Zhang et al. Firstly, we propose the setting of the reward-free exploration with plug-in solver, which is a more general setting that covers the standard reward-free exploration studied in the previous results including Zhang et al. This setting is also more difficult than that studied in Zhang et al., since the agent can only observe the transition estimation in the planning phase and our results holds for any possible planning algorithms. Secondly, our sample complexity bounds are tighter than theirs by a factor of $H^2$ (Note that they study time-homogeneous MDPs setting while we focus on time-inhomogeneous case. Our bound can be further improved by a factor of $H$ when transformed to their setting). Roughly speaking, this is achieved by setting the exploration-driven reward $R_{k,h}(s,a) = u_{k,h}(s,a)$ instead of $u_{k,h}(s,a)/H$ and a more careful analysis to bound the model estimation error in the planning phase. Thirdly, our lower bound is tighter than theirs in the dependence on $d$, which indicates that our sample complexity bound is statistically optimal except for logarithmic factors. We have added more discussion in the related work section.
>
> Regarding the statements and the typos: Thanks for the comments. We have revised the paper accordingly.

---

> > ### Comment · Reviewer_ZcuD · 2021-11-24
> > **Reply to Author's Response**
> >
> > Thanks for the response and the efforts in the updated draft. After reading the revised paper, I have some new comments and hope the authors could address them.
> >
> > Comments:
> >
> > 1. Thanks for the response about the model-based solver, dependence on B, and implementation details. In particular, I have seen some of them are involved in Appendix A.2. However, I would hope all these discussions could be included in the paper. From my perspective, these discussions are valuable for general readers.
> > 2. There are a few typos or confusing parts in Appendix A.2.
> >
> > - "For the case of ﬁnite state space ($\mathcal{S} \leq \infty$）“
> >
> > There is a typo: $\mathcal{S}$ is a set and you should use $|\mathcal{S}|$ (or your notation $S$)  here.
> >
> > - where f = $(f(S_1), \cdots, f(S_{|S|}))^{\top}$
> >
> > There is a typo: f should be $\boldsymbol{\mathrm{f}}$ as in (13).
> >
> > - The optimization problem in line 8 of Algorithm 1 is to ﬁnd parameter satisfying several constraints.
> >
> > There is a typo: line 8 -> line 19.
> >
> > - "The above problem can be regarded as a quadratic programming problem"
> >
> > It seems that $\forall s, a$ are missing in the constraints of this quadratic programming problem.
> >
> >
> >
> > 3. Thanks for the lower bound argument in Appendix C. To my understanding, this lower bound proof (i.e., the proof of Theorem 3) is based on the argument that reward-free exploration is harder than the standard online exploration, which seems plausible. However, I do not believe this is a good way of proving the lower bound. To my understanding, the reward-free exploration allows that the reward function depends on the collected dataset, rather than being fixed. To this end, the lower bound proof of and standard online exploration could be totally different.
> >
> > I elaborate on possible flaws in the proof of Theorem 3 as follows and I apologize if my understanding is wrong. For typical minimax lower bound, the poor performance on hard instances is "algorithm-dependent". Concretely, the hard instances in Theorem 4 work for the class of the standard online exploration algorithms, which know the reward function in advance. Furthermore, these instances are not directly applicable for the class of reward-free exploration algorithms, which does not know the reward function during the exploration phase. These two algorithm classes are different and it is not clear whether there exists one-to-one mapping. Therefore, the contradiction argument (i.e., "Then we can use this algorithm to solve the online exploration problem by simply ignoring the information ...") is not convincing. To build a solid connection or make a reasonable reduction, one-to-one translation between two settings (i.e., problem instances and algorithm classes) is necessary.
> >
> >  In addition, I have to clarify that Theorem 4 seems true for me. It is a high probability bound based on Zhou et al. (2020a), i.e., the regret to sample complexity translation. A typical translation would introduce a polynomial dependence on $\delta$. For example, $1/\delta^2$ as in (Menard et al., 2020). However, the sample complexity $K$ in Theorem 4 does not rely on $\delta$, which seems strange. Could you explain this?
> >
> > In summary, the lower bound proof is not convincing for me and a rigorous proof is required.

---

> > > ### Author Response · Authors · 2021-11-24
> > > **Response to Reviewer ZcuD**
> > >
> > > We thank Reviewer ZcuD for the additional comments.
> > >
> > > Regarding the implementation details: Thanks for the suggestions. We agree that it is better to put the discussion to the main pages. However, it is difficult to rearrange the paper during the rebuttal period due to time limit. We plan to revise the paper accordingly in the final version.
> > >
> > > Regarding the reduction to the online exploration setting: Before we explain the reduction techniques, we want to emphasize that the basic idea of our reduction is not to establish a one-to-one mapping of algorithms or hard instances between the two settings, but to show that any reward-free algorithm can by applied as an algorithm to solve the online exploration problem, which indicates that the reward-free exploration problem is at least as hard as the online exploration problem. Compared with the online exploration setting, we understand that designing efficient algorithms with tight sample complexity *upper bound* for reward-free setting requires to tackle additional difficulties that the reward function may depend on the collected dataset, as stated by the reviewer. However, this does not influence our reduction for the *lower bound* to indicate that reward-free is at least as hard as online exploration. Recall that the lower bound in online exploration setting (Theorem 4) states that for any online exploration algorithm, there exists a hard instance $\mathcal{M}$ with transition $P$ and reward $R$, such that the lower bound to find an $\epsilon$-optimal policy for $\mathcal{M}$ is at least $\Omega(d^2H^3/\epsilon^2)$. To prove the reduction, we construct such an online exploration algorithms by calling a reward-free algorithm as subroutines: For any reward-free algorithm $\mathcal{ALG}$, the online exploration algorithm $\mathcal{AL}G_1$ firstly calls the exploration algorithm of $\mathcal{ALG}$ to interact with the environment, and then calls the planning algorithm of $\mathcal{ALG}$ to calculate and return a policy. By the lower bound for online exploration setting (Theorem 4), for any $\mathcal{AL}G_1$ constructed by $\mathcal{ALG}$, there exists a hard instance $M_{\mathcal{ALG}}$ with transition $P_{\mathcal{ALG}}$ and reward $R_{\mathcal{ALG}}$ such that the agent needs to colect at least $\Omega(d^2H^3/\epsilon^2)$ trajectories to finally return an $\epsilon$-optimal policy. The above statement can be restated in the following way: For any algorithm $\mathcal{ALG}$, suppose $\mathcal{ALG}$ interacts with the MDP instance with the same transition $P_{\mathcal{ALG}}$ during the exploration phase. Then to return a $\epsilon$-optimal policy for the reward $R_{\mathcal{ALG}}$ in the planning phase, the number of episodes collected in the exploration phase needs to be at least $\Omega(d^2H^3/\epsilon^2)$, which leads to the lower bound in Theorem 3. We hope the above discussion clearly explains the idea behind the reduction, and would welcome more discussion for anything still confusing.
> > >
> > > Regarding the dependence on $\delta$: Our current lower bound is derived from a lower bound of the expected regret. Hence the dependence of $\delta$ is only $\Omega(1)$ for constant $\delta$ bounded away from $1$.
> > > In particular, the exact requirement of $\delta$ is set in Eqn 207. Obtaining a stronger lower bound would require a stronger information theoretical argument instead of a reduction (to the expected regret) argument.
> > > Since we are focusing on the $d$ dependence, we leave this question as a future direction. Thanks for pointing out this problem. We will clarify this more in the final version of the paper.

---

> > > > ### Comment · Reviewer_ZcuD · 2021-11-25
> > > > **My Concern is Addressed**
> > > >
> > > > Thanks for the explanation. Now I am convinced in the proof of the lower bound. One more side comment: this lower bound cannot tell us what's hard MDPs for reward-free exploration with function approximation. Thus, it is still valuable to derive the lower bound with common techniques in the future. Since my concern is well addressed, I raise my score from 5 to 6.

---

### Official Review · Reviewer_VReK · 2021-11-03

**Correctness:** 3
**Technical Novelty And Significance:** 3
**Empirical Novelty And Significance:** Not applicable
**Recommendation:** 8
**Confidence:** 2

**Main Review:**

# Strength

- S1. The theoretical results are strong and interesting.
- S2. The paper did a good job such that readers can grasp the high level proof idea.

# Weakness

- W1. There are some typos. Also, there are some arguments and proofs that I can't quite follow.

# Comments and Questions

- C1. In abstract, I think it would be better to replace "interactions" in "taking $\mathcal{O} (d^2 H^3 / \varepsilon^2)$ interactions..." with episodes. I imagine the number of steps $KH$ by "interactions".
- C2. How do you solve $\max\_{V \in \mathcal{V}} \| \phi\_V (s, a) \|\_{\Lambda\_{k, h}^{-1}}$ for $V$ in practice? Is it possible to obtain the optimal $V$ analytically?
- C3. In Page 5, it is said that "By the reduction from the tabular reward-free RL problem ... the linear mixture model setting is $\Omega (\frac{d^2 H^3}{\varepsilon^2})$". It is not unclear for me how to get $d^2$ rather than $d$. Would you explain it a bit more? Also, it would be nice if $H^3$ dependence is formally derived.
- C4. In the equation at the bottom of Page 4, $\beta$ is unnecessary.
- C5. In Page 5, $\phi\_{t, h} (s_h, a_h)$ should be defined as argmax rather than max, I think.
- C6. In Eqn 7, I think $P$ in $u\_{1, k, h}^{\pi, P}$ should be $\widetilde{P}$.
- C7. In Eqn 15, don't you need $B$ on RHS? Is there any assumption on $H$, $d$ etc so that $B$ disappears?
- C8. In Eqn 21, LHS should be $W_h ( \{ R_h \})$.
- C9. In Eqn 29, is $\widetilde{V}_1^{\pi}$ defined anywhere? I can infer its meaning, though.
- C10. In Eqn 49, $H$ seems to be missing in $\log$?
- C11. In Lemma 9, "Under event $\Lambda_1$" should be "Under event $\mathcal{E}_1$"?
- C12. In the definition of the noise $\eta_k$ below Eqn 90, $\theta_h^*$ should be $\theta_h$?
- C13. In Eqn 91 and 92, don't you need $B$?
- C14. In Lemma 16, $\theta_\star$ above Eqn 199 should be $\theta\_\*$, I think. Also in Eqn 199, $\widehat{\theta}\_t$ is undefined, and $\theta$ should be $\theta\_\*$.
- C15. There are some places where $\star$ is used instead of $*$.
- C16. Maybe it is nice to have a list of notations. There are too many $V$ variants, and I needed to look for their definitions.

**Summary Of The Paper:**

The paper propose and analyze a model-based algorithm for the reward-free exploration setting. The analysis shows that the proposed algorithm is (nearly) minimax optimal. The proposed algorithm is agnostic of the planning algorithm that is used to solve an MDP constructed with the estimated model. This property is of independent interest because it allows the computation of the value of the policies, and the estimated model can be used for any down-stream tasks.

**Summary Of The Review:**

I think the results are strong and interesting. There are many typos in equations. While typos are not serious but needs revision. I found some parts of the proofs that I don't quite follow (lacking $B$) and undefined notations.

---

> ### Author Response · Authors · 2021-11-20
> **Response to Reviewer 1**
>
> We thank Reviewer 1 for the positive assessment and insightful comments.
>
> Regarding the optimization problem in Algorithm 1: This problem can be relaxed as a linear programming problem, which can be solved by standard optimization approaches. However, the sample complexity will be worse by a factor of $d$ if we solve the relaxed problem as an approximation. We add more discussion on the optimization issue in Appendix A.2.
>
> Regarding the lower bound: Thanks for pointing out this issue. We understand that the direct reduction cannot lead to the tight lower bound in linear setting. Therefore, we propose a new lower bound of $\Omega(d^2H^3/\epsilon^2)$ (Theorem 3) for the reward-free exploration setting. We achieve this by connecting the sample complexity lower bound with the regret lower bound of certain constructed learning algorithms in the standard online exploration setting. Please see Appendix C for the detailed discussion.
>
> Regarding the dependence on $B$: We have included the explicit dependence on $B$ in the algorithm, the theorem and the proof. Our sample complexity has only logarithmic dependence on $B$.
>
> Regarding the notation table and typos: Thanks for the detailed review. We have added notation tables in the appendix, and fixed typos accordingly.

---

> > ### Comment · Reviewer_VReK · 2021-11-22
> > **Re: Response to Reviewer 1**
> >
> > Thanks a lot for the revision and deriving a new lower bound. The new lower bound is also a nice contribution, I think.

---

### Author Response · Authors · 2021-11-20
**Revision**

We thank all reviewers for the insightful comments and helpful suggestions. We have revised the paper accordingly.
- We propose a lower bound (Theorem 3) in Appendix C, which indicates that our upper bound is minimax optimal except for logarithmic factors.
- We explicitly explain the dependence on $B$ in the algorithm, the theorem and the proof. Our sample complexity has only logarithmic dependence on $B$.
- We discuss the implementation details in Appendix A.2.
- We add more references and compare our work with Zhang et al. in the related work section.
- We discuss in detail how our techniques can help to improve the sample complexity by a factor of $H^2$ in the setting of reward-free exploration in linear MDP (Appendix D).
- We add notation tables in Appendix, and fix typos.

---

### Decision · Program_Chairs · 2022-01-20

**Decision:**

Accept (Spotlight)

**Comment:**

This paper addresses the reward-free exploration problem with function approximation under linear mixture MDP assumption. The analysis shows that the proposed algorithm is (nearly) minimax optimal.  The proposed approach can work with any planning solver to provide an ($\epsilon + \epsilon_{opt}$)-optimal policy for any reward function.

After reading the authors' feedback and discussing their concerns, the reviewers agree that the contributions in this paper are valuable and that this paper deserves publication.
I encourage the authors to follow the reviewers' suggestions as they will prepare the camera-ready version.